



Nonlinear Processes
in Geophysics

# Fast hybrid tempered ensemble transform filter formulation for Bayesian elliptical problems via Sinkhorn approximation

**Sangeetika Ruchi**[1], **Svetlana Dubinkina**[1,3], **and Jana de Wiljes**[2]

[1]Centrum Wiskunde & Informatica, P.O. Box 94079, 1098 XG Amsterdam, the Netherlands
[2]Institute of Mathematics, University of Potsdam, Karl-Liebknecht-Str. 24/25, 14476 Potsdam, Germany
[3]Department of Mathematics, VU Amsterdam, De Boelelaan 1081, 1081 HV, Amsterdam, the Netherlands

**Correspondence:** Jana de Wiljes (wiljes@uni-potsdam.de)

**Abstract.** Identification of unknown parameters on the basis of partial and noisy data is a challenging task, in particular in high dimensional and non-linear settings. Gaussian approximations to the problem, such as ensemble Kalman inversion, tend to be robust and computationally cheap and often produce astonishingly accurate estimations despite the simplifying underlying assumptions. Yet there is a lot of room for improvement, specifically regarding a correct approximation of a non-Gaussian posterior distribution. The tempered ensemble transform particle filter is an adaptive Sequential Monte Carlo (SMC) method, whereby resampling is based on optimal transport mapping. Unlike ensemble Kalman inversion, it does not require any assumptions regarding the posterior distribution and hence has shown to provide promising results for non-linear non-Gaussian inverse problems. However, the improved accuracy comes with the price of much higher computational complexity, and the method is not as robust as ensemble Kalman inversion in high dimensional problems. In this work, we add an entropy-inspired regularisation factor to the underlying optimal transport problem that allows the high computational cost to be considerably reduced via Sinkhorn iterations. Further, the robustness of the method is increased via an ensemble Kalman inversion proposal step before each update of the samples, which is also referred to as a hybrid approach. The promising performance of the introduced method is numerically verified by testing it on a steady-state single-phase Darcy flow model with two different permeability configurations. The results are compared to the output of ensemble Kalman inversion, and Markov chain Monte Carlo methods results are computed as a benchmark.

## 1 Introduction

If a solution of a considered partial differential equation (PDE) is highly sensitive to its parameters, accurate estimation of the parameters and their uncertainties is essential to obtain a correct approximation of the solution. Partial observations of the solution are then used to infer uncertain parameters by solving a PDE-constrained inverse problem. For instance, one can approach such problems via methods induced by Bayes' formula (Stuart, 2010). More specifically, the posterior probability density of the parameters given the data is then computed on the basis of a prior probability density and a likelihood, which is the conditional probability density associated with the given noisy observations. The well-posedness of an inverse problem and convergence to the true posterior in the limit of observational noise going to zero were proven for different priors and under assumptions on the parameter-to-observation map by Dashti and Stuart (2017), for example.

When aiming at practical applications as in oil reservoir management (Lorentzen et al., 2020) and meteorology (Houtekamer and Zhang, 2016), for example, the posterior is approximated by means of a finite set of samples. Markov chain Monte Carlo (MCMC) methods approximate the posterior with a chain of samples – a sequential update of samples according to the posterior (Robert and Casella, 2004; Rosenthal, 2009; Hoang et al., 2013). Typically, MCMC methods provide highly correlated samples. Therefore, in order to sample the posterior correctly, MCMC requires a long chain, especially in the case of a multi-modal or a peaked distribution. A peaked posterior is associated

with very accurate observations. Therefore, unless a speed-up is introduced in a MCMC chain (e.g. Cotter et al., 2013), MCMC is impractical for computationally expensive PDE models.

Adaptive Sequential Monte Carlo (SMC) methods are different approaches to approximate the posterior with an *ensemble* of samples by computing their probability (e.g, Vergé et al., 2015). Adaptive intermediate probability measures are introduced between the prior measure and the posterior measure to improve upon method divergence due to the curse of dimensionality following Del Moral et al. (2006) and Neal (2001). Moreover, sampling from an invariant Markov kernel with the target intermediate measure and the reference prior measure improves upon ensemble diversity due to parameters' stationarity, as shown by Beskos et al. (2015). However, when parameter space is high dimensional, adaptive SMC requires computationally prohibitive ensemble sizes unless we approximate only the first two moments (e.g. Iglesias et al., 2018) or we sample highly correlated samples (Ruchi et al., 2019).

Ensemble Kalman inversion (EKI) approximates primarily the first two moments of the posterior, which makes it computationally attractive for estimating high dimensional parameters (Iglesias et al., 2014). For linear problems, Blömker et al. (2019) showed well-posedness and convergence of the EKI for a fixed ensemble size and without any assumptions of Gaussianity. However for non-linear problems, it has been shown by Oliver et al. (1996), Bardsley et al. (2014), Ernst et al. (2015), Liu et al. (2017) and Le Gland et al. (2011) that the EKI approximation is not consistent with the Bayesian approximation.

We note that the EKI is an iterative ensemble smoother (Evensen, 2018). Iterative ensemble smoothers for inverse problems introduce a trivial artificial dynamics to the unknown static parameter and iteratively update an estimation of the parameter. Then the parameter-dependent model variables are recomputed using a forward model with a parameter estimation. Examples of iterative ensemble smoothers are ensemble randomised maximum likelihood (Chen and Oliver, 2012), multiple data assimilation (Emerick and Reynolds, 2013) and randomise-then-optimise (Bardsley et al., 2014).

As an alternative ansatz one can employ optimal transport resampling that lies at the heart of the ensemble transform particle filter (ETPF) proposed by Reich (2013). An optimal transport map between two consecutive probability measures provides a direct sample-to-sample map with maximised sample correlation. Along the lines of an adaptive SMC approach, a probability measure is described via the importance weights, and the deterministic mapping replaces the traditional resampling step. A so-called tempered ensemble transform particle filter (TETPF) was proposed by Ruchi et al. (2019). Note that this ansatz does not require any distributional assumption for the posterior, and it was shown by Ruchi et al. (2019) that the TETPF provides encouraging

results for non-linear high dimensional PDE-constrained inverse problems. However, the computational cost of solving an optimal transport problem in each iteration is considerably high.

In this work we address two issues that have arisen in the context of the TETPF: (i) the immense computational costs of solving the associated optimal transport problem and (ii) the lack of robustness of the TETPF with respect to high dimensional problems. More specifically, the performance of ETPF has been found to be highly dependent on the initial guess. Although tempering restrains any sharp fail in the importance sampling step due to a poor initial ensemble selection, the number of required intermediate steps and the efficiency of ETPF still depend on the initialisation. The lack of robustness in high dimensions can be addressed via a hybrid approach that combines a Gaussian approximation with a particle filter approximation (e.g. Santitissadeekorn and Jones, 2015). Different algorithms are created by Frei and Künsch (2013) and Stordal et al. (2011), for example. In this paper, we adapt a hybrid approach of Chustagulprom et al. (2016) that uses the EKI as a proposal step for the ETPF with a tuning parameter. Furthermore, it is well established that the computational complexity of solving an optimal transport problem can be significantly reduced via a Sinkhorn approximation by Cuturi (2013). This ansatz has been implemented for the ETPF by Acevedo et al. (2017).

Along the lines of Chustagulprom et al. (2016) and de Wiljes et al. (2020), we propose a tempered ensemble transform particle filter with Sinkhorn approximation (TESPF) and a tempered hybrid approach.

The remainder of the paper is organised as follows: in Sect. 2, the inverse problem setting is presented. There we describe the tempered ensemble transform particle filter (TETPF) proposed by Ruchi et al. (2019). Furthermore, we introduce the tempered ensemble transform particle filter with Sinkhorn approximation (TESPF), a tempered hybrid approach that combines the EKI and the TETPF (hybrid EKI–TETPF), and a tempered hybrid approach that combines the EKI and the TESPF (hybrid EKI–TESPF). We provide pseudocodes of all the presented techniques in Appendix A and corresponding computational complexities in Appendix B. In Sect. 3, we apply the adaptive SMC methods to an inverse problem of inferring high dimensional permeability parameters for a steady-state single-phase Darcy flow model. Permeability is parameterised following Ruchi et al. (2019), whereby one configuration of parameterisation leads to Gaussian posteriors, while another one leads to non-Gaussian posteriors. Finally, we draw conclusions in Sect. 4.

## 2    Bayesian inverse problem

We assume $\boldsymbol{u} \in \tilde{\mathcal{U}} \subset \mathbb{R}^n$ is a random variable that is related to partially observable quantities $\boldsymbol{y} \in \mathcal{Y} \subset \mathbb{R}^\kappa$ by a non-linear

forward operator $G : \tilde{\mathcal{U}} \to \mathcal{Y}$, namely

$$\boldsymbol{y} = G(\boldsymbol{u}).$$

Further, $\boldsymbol{y}_{\mathrm{obs}} \in \mathcal{Y}$ denotes a noisy observation of $\boldsymbol{y}$, i.e.

$$\boldsymbol{y}_{\mathrm{obs}} = \boldsymbol{y} + \boldsymbol{\eta},$$

where $\boldsymbol{\eta} \sim \mathcal{N}(\boldsymbol{0}, \mathbf{R})$ and $\mathcal{N}(\boldsymbol{0}, \mathbf{R})$ is a Gaussian distribution with zero mean and $\mathbf{R}$ covariance matrix. The aim is to determine or approximate the posterior measure $\mu(\boldsymbol{u})$ conditioned on observations $\boldsymbol{y}_{\mathrm{obs}}$ and given a prior measure $\mu_0(\boldsymbol{u})$, which is referred to as a Bayesian inverse problem. The posterior
measure is absolutely continuous with respect to the prior, i.e.

$$\frac{\mathrm{d}\mu}{\mathrm{d}\mu_0}(\boldsymbol{u}) \propto g(\boldsymbol{u}; \boldsymbol{y}_{\mathrm{obs}}), \tag{1}$$

where $\propto$ is up to a constant of normalisation, and $g$ is referred to as the likelihood and depends on the forward operator $G$.
The Gaussian observation noise of the observation $\boldsymbol{y}_{\mathrm{obs}}$ implies

$$g(\boldsymbol{u}; \boldsymbol{y}_{\mathrm{obs}}) = \exp\left[ -\frac{1}{2}(G(\boldsymbol{u}) - \boldsymbol{y}_{\mathrm{obs}})' \mathbf{R}^{-1} (G(\boldsymbol{u}) - \boldsymbol{y}_{\mathrm{obs}}) \right], \quad (2)$$

where $'$ denotes the transpose. In the following we will introduce a range of methods that can be employed to estimate
solutions to the presented inverse problem under the overarching mantel of tempered Sequential Monte Carlo filters. Alongside these methods we will also propose several important add-on tools required to achieve feasibility and higher accuracy in high dimensional non-linear settings.

## 2.1 Tempered Sequential Monte Carlo

We consider Sequential Monte Carlo (SMC) methods that approximate the posterior measure $\mu(\boldsymbol{u})$ via an empirical measure

$$\mu^M(\boldsymbol{u}) = \sum_{i=1}^{M} w_i \delta_{\boldsymbol{u}_i}(\boldsymbol{u}).$$

Here, $\delta$ is the Dirac function, and the importance weights for the approximation of $\mu$ are

$$w_i = \frac{g(\boldsymbol{u}_i; \boldsymbol{y}_{\mathrm{obs}})}{\sum_{j=1}^{M} g(\boldsymbol{u}_j; \boldsymbol{y}_{\mathrm{obs}})}.$$

An ensemble $\mathcal{U} = \{\boldsymbol{u}_1, \ldots, \boldsymbol{u}_M\} \subset \tilde{\mathcal{U}}$ consists of $M$ realisations $\boldsymbol{u}_i \in \mathbb{R}^n$ of a random variable $\boldsymbol{u}$ that are independent
and identically distributed according to $\boldsymbol{u}_i \sim \mu_0$.

When an easy-to-sample ensemble from the prior $\mu_0$ does not approximate the complex posterior $\mu$ well, only a few weights $w_i$ have significant value, resulting in a degenerative approximation of the posterior measure. Potential reasons for

this effect are high dimensionality of the uncertain parame-
ter, a large number of observations, or lack of accuracy of the observations. An existing solution to a degenerative approximation is an iterative approach based on tempering by Del Moral et al. (2006) or annealing by Neal (2001). The underlying idea is to introduce $T$ intermediate artificial mea-
sures $\{\mu_t\}_{t=0}^{T}$ between $\mu_0$ and $\mu_t = \mu$. These measures are bridged by introducing $T$ tempering parameters $\{\phi_t\}_{t=1}^{T}$ that satisfy $0 = \phi_0 < \phi_1 < \ldots < \phi_t = 1$ `TS1`. An intermediate measure $\mu_t$ is defined as a probability measure that has density proportional to $g(\boldsymbol{u})$ with respect to the previous measure
$\mu_{t-1}$:

$$\frac{\mathrm{d}\mu_t}{\mathrm{d}\mu_{t-1}}(\boldsymbol{u}) \propto g(\boldsymbol{u}; \boldsymbol{y}_{\mathrm{obs}})^{(\phi_t - \phi_{t-1})}.$$

Along the lines of Iglesias (2016) the tempering parameter $\phi_t$ is chosen such that the effective sample size (ESS),

$$\mathrm{ESS}_t(\phi) = \frac{\left( \sum_{i=1}^{M} w_{t,i} \right)^2}{\sum_{i=1}^{M} w_{t,i}^2}, \tag*{55}$$

with

$$w_{t,i} = \frac{g(\boldsymbol{u}_{t-1,i}; \boldsymbol{y}_{\mathrm{obs}})^{(\phi_t - \phi_{t-1})}}{\sum_{j=1}^{M} g(\boldsymbol{u}_{t-1,j}; \boldsymbol{y}_{\mathrm{obs}})^{(\phi_t - \phi_{t-1})}}, \tag{3}$$

does not drop below a certain threshold $1 < M_{\mathrm{thresh}} < M$. Then, an approximation of the posterior measure $\mu_t$ is

$$\mu_t^M(\boldsymbol{u}) = \sum_{i=1}^{M} w_{t,i} \delta_{\boldsymbol{u}_{t-1,i}}(\boldsymbol{u}). \tag{4}$$

A bisection algorithm on the interval $(\phi_{t-1}, 1]$ is employed to find $\phi_t$. If $\mathrm{ESS}_t > M_{\mathrm{thresh}}$, we set $\phi_t = 1$ `TS2`, which implies that no further tempering is required.

The choice of ESS to define a tempering parameter is supported by results of Beskos et al. (2014) on the stability of
a tempered SMC method in terms of the ESS. Moreover, for a Gaussian probability density approximated by importance sampling, Agapiou et al. (2017) showed that ESS is related to the second moment of the Radon–Nikodym derivative (Eq. 1).
The SMC method with importance sampling (Eq. 4) does not change the sample $\{\boldsymbol{u}_{t-1,i}\}_{i=1}^{M}$, which leads to the method collapse due to a finite ensemble size. Therefore each tempering iteration $t$ needs to be supplied with resampling. Resampling provides a new ensemble $\{\tilde{\boldsymbol{u}}_{t,i}\}_{i=1}^{M}$ that approximates
the measure $\mu_t$. We will discuss different resampling techniques in Sect. 2.3.

## 2.2 Mutation

Due to the stationarity of the parameters, SMC methods require ensemble perturbation. In the framework of parti-
cle filtering for dynamical systems, ensemble perturbation is

achieved by rejuvenation, when ensemble members of the posterior measure are perturbed with a random noise sampled from a Gaussian distribution with zero mean and a covariance matrix of the prior measure. The covariance matrix of the ensemble is inflated, and no acceptance step is performed due to the associated high computational costs for a dynamical system.

Since we consider a static inverse problem, for ensemble perturbation we employ a Metropolis–Hastings method (thus we mutate samples) but with a proposal that speeds up the MCMC method for estimating a high dimensional parameter. Namely, we use the ensemble mutation of Cotter et al. (2013) with the target measure $\mu_t$ and the reference measure $\mu_0$. The mutation phase is initialised at $\boldsymbol{v}_{0,i} = \tilde{\boldsymbol{u}}_{t,i}$, and at the final inner iteration $\tau_{\max}$, we assign $\boldsymbol{u}_{t,i} = \boldsymbol{v}_{\tau_{\max},i}$ for $i = 1, \ldots, M$.

For a Gaussian prior we use the preconditioned Crank–Nicolson MCMC (pcn-MCMC) method:

$$
\begin{aligned}
\boldsymbol{v}_i^{\text{prop}} &= \sqrt{1-\theta^2}\,\boldsymbol{v}_{\tau,i} + (1 - \sqrt{1-\theta^2})\boldsymbol{m} \\
&\quad + \theta\boldsymbol{\xi}_{\tau,i} \text{ for } i = 1, \ldots, M.
\end{aligned} \tag{5}
$$

Here, $\boldsymbol{m}$ is the mean of the Gaussian prior measure $\mu_0$, and $\{\boldsymbol{\xi}_{\tau,i}\}_{i=1}^M$ are from a Gaussian distribution with zero mean and a covariance matrix of the Gaussian prior measure $\mu_0$.

For a uniform prior $U[a, b]$ we use the following random walk:

$$
\boldsymbol{v}_i^{\text{prop}} = \boldsymbol{v}_{\tau,i} + \boldsymbol{\xi}_{\tau,i} \text{ for } i = 1, \ldots, M. \tag{6}
$$

Here $\{\boldsymbol{\xi}_{\tau,i}\}_{i=1}^M \sim U[a-b, b-a]$ and $\{\boldsymbol{v}_i^{\text{prop}}\}_{i=1}^M$ are projected onto the $[a, b]$ interval if necessary. Then the ensemble at the inner iteration $\tau + 1$ is

$$
\begin{aligned}
\boldsymbol{v}_{\tau+1,i} &= \boldsymbol{v}_i^{\text{prop}} \text{ with the probability} \\
&\quad \rho(\boldsymbol{v}_i^{\text{prop}}, \boldsymbol{u}_{t-1,i}) \text{ for } i = 1, \ldots, M;
\end{aligned} \tag{7}
$$

$$
\begin{aligned}
\boldsymbol{v}_{\tau+1,i} &= \boldsymbol{v}_{\tau,i} \text{ with the probability} \\
&\quad 1 - \rho(\boldsymbol{v}_i^{\text{prop}}, \boldsymbol{u}_{t-1,i}) \text{ for } i = 1, \ldots, M.
\end{aligned} \tag{8}
$$

Here $\boldsymbol{v}_i^{\text{prop}}$ is from Eq. (5) for the Gaussian measure and from Eq. (6) for the uniform measure, and

$$
\rho(\boldsymbol{v}_i^{\text{prop}}, \boldsymbol{u}_{t-1,i}) = \min\left\{1, \frac{g(\boldsymbol{v}_i^{\text{prop}}; \boldsymbol{y}_{\text{obs}})^{\phi_t}}{g(\boldsymbol{u}_{t-1,i}; \boldsymbol{y}_{\text{obs}})^{\phi_t}}\right\}.
$$

The scalar $\theta \in (0, 1]$ in Eq. (5) controls the performance of the Markov chain. Small values of $\theta$ lead to high acceptance rates but poor mixing. Roberts and Rosenthal (2001) showed that for high dimensional problems it is optimal to choose $\theta$ such that the acceptance rate is in between 20 % and 30 % by the last tempering iteration $T$. Cotter et al. (2013) proved that under some assumptions this mutation produces a Markov kernel with an invariant measure $\mu_t$.

*Computational complexity.* In each tempering iteration $t$ the computational complexity of the pcn-MCMC mutation is $\mathcal{O}(\tau_{\max} M \mathcal{C})$, where $\mathcal{C}$ is the computational cost of the forward model $G$. For the pseudocode of the pcn-MCMC mutation, please refer to Algorithm 1 in Appendix A. Note that the computational complexity is not affected by the length of $\boldsymbol{u}$, which is a very desirable property in high dimensions as shown by Cotter et al. (2013) and Hairer et al. (2014).

## 2.3 Resampling phase

As we have already mentioned in Sect. 2.1, an adaptive SMC method with importance sampling needs to be supplied with resampling at each tempering iteration $t$. We consider a resampling method based on optimal transport mapping proposed by Reich (2013).

### 2.3.1 Optimal transformation

The origin of the optimal transport theory lies in finding an optimal way of redistributing mass which was first formulated by Monge (1781). Given a distribution of matter, e.g. a pile of sand, the underlying question is how to reshape the matter into another form such that the work done is minimal. A century and a half later the original problem was rewritten by Kantorovich (1942) in a statistical framework that allowed it to be tackled. Due to these contributions it was later named the Monge–Kantorovich minimisation problem. The reader is also referred to Peyré and Cuturi (2019) for a comprehensible overview.

Let us consider a scenario whereby the initial distribution of matter is represented by a probability measure $\mu$ on the measurable space $\mathcal{U}$, that has to be moved and rearranged according to a given new distribution $\nu$, defined on the measurable space $\tilde{\mathcal{U}}$. In order to describe the link between the two probability measures $\mu$ and $\nu$ and to minimise a predefined cost associated with the *transportation*, one aims to find a joint measure on $\mathcal{U} \times \tilde{\mathcal{U}}$ that is a solution to

$$
\inf\left\{ \int_{\mathcal{U} \times \tilde{\mathcal{U}}} c(\boldsymbol{u}, \tilde{\boldsymbol{u}}) \mathrm{d}\omega(\boldsymbol{u}, \tilde{\boldsymbol{u}}) : \omega \in \prod(\mu, \nu) \right\}, \tag{9}
$$

where the minimum is computed over all joint probability measures $\omega$ on $\mathcal{U} \times \tilde{\mathcal{U}}$, denoted $\prod(\mu, \nu)$, with marginals $\mu$ and $\nu$, and $c(\boldsymbol{u}, \tilde{\boldsymbol{u}})$ is a transport cost function on $(\boldsymbol{u}, \tilde{\boldsymbol{u}}) \in \mathcal{U} \times \tilde{\mathcal{U}}$. The joint measures achieving the infinum are called optimal transport plans.

Let $\mu$ and $\nu$ be two measures on a measurable space $(\Omega, \mathcal{F})$ such that $\mu$ is the law of random variable $U : \Omega \to \mathcal{U}$ and $\nu$ is the law of random variable $\tilde{U} : \Omega \to \tilde{\mathcal{U}}$. Then a coupling of $(\mu, \nu)$ consists of a pair $(U, \tilde{U})$. Note that couplings always exist; an example is the trivial coupling in which the random variables $U$ and $\tilde{U}$ are independent. A coupling is called deterministic if there is a measurable function $\Psi_M : \mathcal{U} \to \tilde{\mathcal{U}}$ such that $\tilde{U} = \Psi_M(U)$, and $\Psi_M$ is called a transport map. Unlike general couplings, deterministic couplings do not always exist. On the other hand there may be

infinitely many deterministic couplings. One famous variant of Eq. (9), whereby the optimal coupling is known to be a deterministic coupling, is given by

$$\omega^* = \arg\inf \left\{ \int_{\mathcal{U} \times \tilde{\mathcal{U}}} \|\boldsymbol{u} - \tilde{\boldsymbol{u}}\|^2 \mathrm{d}\omega(\boldsymbol{u}, \tilde{\boldsymbol{u}}) : \omega \in \prod(\mu, \nu) \right\}. \quad (10)$$

The aim of the resampling step is to obtain equally probable samples. Therefore, in resampling based on optimal transport of Reich (2013), the Monge–Kantorovich minimisation problem (Eq. 10) is considered for the current posterior measure $\mu_t^M(\boldsymbol{u})$ given by its samples approximation (Eq. 4) and a uniform measure (here the weights in the sample approximation are set to $1/M$). The discretised objective functional of the associate optimal transport problem is given by

$$J(\mathbf{S}) := \sum_{i,j=1}^{M} s_{ij} \|\boldsymbol{u}_{t-1,i} - \boldsymbol{u}_{t-1,j}\|^2,$$

subject to $s_{ij} > 0$ and constraints

$$\sum_{i=1}^{M} s_{ij} = \frac{1}{M}, j = 1, \ldots M; \sum_{j=1}^{M} s_{ij} = w_{t,i}, i = 1, \ldots M,$$

where matrix $\mathbf{S}$ describes a joint probability measure under the assumption that the state space is finite. Then samples $\{\tilde{\boldsymbol{u}}_{t,i}\}_{i=1}^{M}$ are obtained by a deterministic linear transform, i.e.

$$\tilde{\boldsymbol{u}}_{t,j} := M \sum_{i=1}^{M} \boldsymbol{u}_{t-1,i} s_{ij} \text{ for } j = 1, \ldots, M. \quad (11)$$

Reich (2013) showed weak convergence of the deterministic optimal transformation (Eq. 11) to a solution of the Monge–Kantorovich problem (Eq. 9) as $M \to \infty$.

*Computational complexity.* The computational complexity of solving the optimal transport problem with an efficient earth mover's distance algorithm such as FastEMD of Pele and Werman (2009) is of order $\mathcal{O}(M^3 \log M)$. Consequently the computational complexity of the adaptive tempering SMC with optimal transport resampling (TETPF) is $\mathcal{O}[T(M\mathcal{C} + M^3 \log M + \tau_{\max} M\mathcal{C})]$, where $T$ is the number of tempering iterations, $\tau_{\max}$ is the number of pcn-MCMC inner iterations and $\mathcal{C}$ is the computational cost of a forward model $G$. For the pseudocode of the TETPF, please refer to Algorithm 2 in Appendix A.

### 2.3.2 Sinkhorn approximation

As discussed above, solving the optimal transport problem has a computational complexity of $\mathcal{O} = M^3 \log(M)$ in every iteration of the tempering procedure. Thus the TETPF becomes very expensive for large $M$. On the other hand an increase in the number of samples directly correlates with an improved accuracy of the estimation. In order to allow for

as many samples as possible, one needs to reduce the associated computational cost of the optimal transport problem. This can be achieved by replacing the optimal transport distance with a Sinkhorn distance and subsequently exploiting the new structure to elude the immense computational time of the EMD (Earth mover's distance) solver, as shown by Cuturi (2013). More precisely the ansatz is built on the fact that the original transport problem has a natural entropic bound that is obtained for $\mathbf{S} = [\frac{1}{M} \boldsymbol{I}_M \boldsymbol{w}^\top]$, where $\boldsymbol{w} = [w_1, \ldots, w_M]$ and $\boldsymbol{I}_M = [1, \ldots, 1] \in \mathbb{R}^M$, which constitutes an independent joint probability. Therefore, one can consider the problem of finding a matrix $\mathbf{S} \in \mathbb{R}^{M \times M}$ that is constrained by an additional lower entropic bound (Sinkhorn distance). This additional constraint can be incorporated via a Lagrange multiplier, which leads to the above regularised form, i.e.

$$J_{\mathrm{SH}}(\mathbf{S}) = \sum_{i,j=1}^{M} \left\{ s_{ij} \|\boldsymbol{u}_{t-1,i} - \boldsymbol{u}_{t-1,j}\|^2 + \frac{1}{\alpha} s_{ij} \log s_{ij} \right\}, \quad (12)$$

where $\alpha > 0$. Due to additional smoothness the minimum of Eq. (12) can be unique and has the form

$$\mathbf{S}^\alpha = \mathbf{diag}(\boldsymbol{b}) \exp\left(-\alpha \mathbf{Z}\right) \mathbf{diag}(\boldsymbol{a}),$$

where $\mathbf{Z}$ is a matrix with entries $z_{ij} = \|\boldsymbol{u}_{t-1,i} - \boldsymbol{u}_{t-1,j}\|^2$ and $\boldsymbol{b}$ and $\boldsymbol{a}$ non-negative vectors determined by employing Sinkhorn's fixed point iteration described by Sinkhorn (1967). We will refer to this approach as the tempered ensemble Sinkhorn particle filter (TESPF).

*Computational complexity.* Solving this regularised optimal transport problem rather than the original transport problem given in Eq. (9) reduces the complexity to $\mathcal{O}(M^2 C(\alpha))$, where $C(\alpha)$ denotes a computational scaling factor that depends on the choice of the regularisation factor $\alpha$. In particular $C(\alpha)$ grows with $\alpha$. Therefore, one needs to balance between reducing computational time and finding a reasonable approximate solution of the original transport problem when choosing a value for $\alpha$. For the pseudocode of the Sinkhorn adaptation of solving the optimal transport problem, please refer to Algorithm 3 in Appendix A. For the pseudocode of the TESPF, please refer to Algorithm 4 in Appendix A.

### 2.4 Ensemble Kalman inversion

For Bayesian inverse problems with Gaussian measures, ensemble Kalman inversion (EKI) is one of the widely used algorithms. The EKI is an adaptive SMC method that approximates primarily the first two statistical moments of a posterior distribution. For a linear forward model, the EKI is optimal in the sense that it minimises the error in the mean (Blömker et al., 2019). For a non-linear forward model, the EKI still provides a good estimation of the posterior (e.g. Iglesias et al., 2018). Here we consider the EKI method of Iglesias et al. (2018), since it is based on the tempering approach.

The intermediate measures $\{\mu_t\}_{t=0}^{T}$ are approximated by Gaussian distributed variables with empirical mean $\boldsymbol{m}_t$ and empirical variance $\mathbf{C}_t$. Empirical mean $\boldsymbol{m}_{t-1}$ and empirical covariance $\mathbf{C}_{t-1}$ are defined in terms of $\{\boldsymbol{u}_{t-1,i}\}_{i=1}^{M}$ as follows:

$$\boldsymbol{m}_{t-1} = \frac{1}{M} \sum_{i=1}^{M} \boldsymbol{u}_{t-1,i},$$

$$\mathbf{C}_{t-1} = \frac{1}{M-1} \sum_{i=1}^{M} (\boldsymbol{u}_{t-1,i} - \boldsymbol{m}_{t-1}) \otimes (\boldsymbol{u}_{t-1,i} - \boldsymbol{m}_{t-1}),$$

where $\otimes$ denotes the Kronecker product. Then the mean and the covariance are updated as

$$\boldsymbol{m}_t = \boldsymbol{m}_{t-1} + \mathbf{C}_{t-1}^{\mathrm{uG}} (\mathbf{C}_{t-1}^{\mathrm{GG}} + \Delta_t \mathbf{R})^{-1} (\boldsymbol{y}_{\mathrm{obs}} - \overline{\boldsymbol{G}}_{t-1}) \text{ and}$$

$$\mathbf{C}_t = \mathbf{C}_{t-1} - \mathbf{C}_{t-1}^{\mathrm{uG}} (\mathbf{C}_{t-1}^{\mathrm{GG}} + \Delta_t \mathbf{R})^{-1} (\mathbf{C}_{t-1}^{\mathrm{uG}})',$$

respectively. Here $'$ denotes the transpose,

$$\mathbf{C}_{t-1}^{\mathrm{uG}} = \frac{1}{M-1} \sum_{i=1}^{M} (\boldsymbol{u}_{t-1,i} - \boldsymbol{m}_{t-1}) \otimes (G(\boldsymbol{u}_{t-1,i}) - \overline{\boldsymbol{G}}_{t-1}),$$

$$\mathbf{C}_{t-1}^{\mathrm{GG}} = \frac{1}{M-1} \sum_{i=1}^{M} [G(\boldsymbol{u}_{t-1,i}) - \overline{\boldsymbol{G}}_{t-1}] \otimes [G(\boldsymbol{u}_{t-1,i}) - \overline{\boldsymbol{G}}_{t-1}],$$

$$\overline{\boldsymbol{G}}_{t-1} = \frac{1}{M} \sum_{i=1}^{M} G(\boldsymbol{u}_{t-1,i}), \text{ and } \Delta_t = \frac{1}{\phi_t - \phi_{t-1}}.$$

We recall that the non-linear forward problem is $\boldsymbol{y} = G(\boldsymbol{u})$, the observation $\boldsymbol{y}_{\mathrm{obs}}$ has a Gaussian observation noise with zero mean and the covariance matrix $\mathbf{R}$ and $\phi_t$ is a temperature associated with the measure $\mu_t$.

Since we are interested in an ensemble approximation of the posterior distribution, we update the ensemble members by

$$\tilde{\boldsymbol{u}}_{t,i} = \boldsymbol{u}_{t-1,i} + \mathbf{C}_{t-1}^{\mathrm{uG}} (\mathbf{C}_{t-1}^{\mathrm{GG}} + \Delta_t \mathbf{R})^{-1} [\boldsymbol{y}_{t,i} - G(\boldsymbol{u}_{t-1,i})]$$
$$\text{for } i = 1, \ldots, M. \tag{13}$$

Here $\boldsymbol{y}_{t,i} = \boldsymbol{y}_{\mathrm{obs}} + \boldsymbol{\eta}_{t,i}$ and $\boldsymbol{\eta}_{t,i} \sim \mathcal{N}(\boldsymbol{0}, \Delta_t \mathbf{R})$ for $i = 1, \ldots, M$.

*Computational complexity.* The computational complexity of solving Eq. (13) is $\mathcal{O}(\kappa^2 n)$, where $n$ is the parameter space dimension, and $\kappa$ is the observation space dimension. Then the computational complexity of the EKI is $\mathcal{O}[T(M\mathcal{C} + \kappa^2 n + \tau_{\max} M\mathcal{C})]$, where $T$ is the number of tempering iterations, $\tau_{\max}$ is the number of pcn-MCMC inner iterations and $\mathcal{C}$ is the computational cost of a forward model $G$. For the pseudocode of the EKI method, please refer to Algorithm 5 in Appendix A.

## 2.5 Hybrid

Despite the underlying Gaussian assumption, the EKI is remarkably robust in non-linear high dimensional settings as opposed to consistent SMC methods such as the TET(S)PF. For many non-linear problems it is desirable to have better uncertainty estimates while maintaining a level of robustness. This can be achieved by factorising the likelihood given by Eq. (2), e.g,

$$g(\boldsymbol{u}; \boldsymbol{y}_{\mathrm{obs}}) = g_1(\boldsymbol{u}; \boldsymbol{y}_{\mathrm{obs}}) \cdot g_2(\boldsymbol{u}; \boldsymbol{y}_{\mathrm{obs}}),$$

where

$$g_1(\boldsymbol{u}; \boldsymbol{y}_{\mathrm{obs}}) = g(\boldsymbol{u}; \boldsymbol{y}_{\mathrm{obs}})^\beta$$
$$= \exp\left[-\frac{1}{2}(G(\boldsymbol{u}) - \boldsymbol{y}_{\mathrm{obs}})'\beta(\mathbf{R})^{-1}(G(\boldsymbol{u}) - \boldsymbol{y}_{\mathrm{obs}})\right] \tag{14}$$

and TS3

$$g_2(\boldsymbol{u}; \boldsymbol{y}_{\mathrm{obs}}) = g(\boldsymbol{u}; \boldsymbol{y}_{\mathrm{obs}})^{(1-\beta)}$$
$$= \exp\left[-\frac{1}{2}(G(\boldsymbol{u}) - \boldsymbol{y}_{\mathrm{obs}})'(1-\beta)[\mathbf{R}]^{-1}\right.$$
$$\left.(G(\boldsymbol{u}) - \boldsymbol{y}_{\mathrm{obs}})\right]. \tag{15}$$

Then TS4 it is possible to alternate between methods with complementing properties such as the EKI and the TET(S)PF updates; e.g. likelihood,

$$\exp\left[-\frac{\beta}{2}(G(\boldsymbol{u}) - \boldsymbol{y}_{\mathrm{obs}})'\mathbf{R}^{-1}(G(\boldsymbol{u}) - \boldsymbol{y}_{\mathrm{obs}})\right]^{(\phi_t - \phi_{t-1})},$$

is used for an EKI update followed by an update with a TET(S)PF on the basis of

$$\exp\left[-\frac{(1-\beta)}{2}(G(\boldsymbol{u}) - \boldsymbol{y}_{\mathrm{obs}})'\mathbf{R}^{-1}(G(\boldsymbol{u}) - \boldsymbol{y}_{\mathrm{obs}})\right]^{(\phi_t - \phi_{t-1})}.$$

Note that $\beta \in [0, 1]$ and should be tuned according to the underlying forward operator. This combination of an approximative Gaussian method and a consistent SMC method has been referred to as hybrid filters in the data assimilation literature[1](Stordal et al., 2011; Frei and Künsch, 2013; Chustagulprom et al., 2016). This ansatz can also be understood as using the EKI as a more elaborate proposal density for the importance sampling step within SMC (e.g. Oliver et al., 1996).

*Computational complexity.* The computational complexity of combining the two algorithms is $\mathcal{O}[T(M\mathcal{C} + \kappa^2 n + M\mathcal{C} + M^3 \log M + \tau_{\max} M\mathcal{C})]$ for the hybrid EKI–TETPF and $\mathcal{O}[T(M\mathcal{C} + \kappa^2 n + M\mathcal{C} + M^2 C(\alpha) + \tau_{\max} M\mathcal{C})]$ for the hybrid EKI–TESPF. For the pseudocode of the hybrid methods, please refer to Algorithm 6 in Appendix A.

## 3 Numerical experiments

We consider a steady-state single-phase Darcy flow model defined over an aquifer of a two-dimensional physical do-

---

[1]Note that the terminology is also used in the context of data assimilation filters combining variational and sequential approaches.

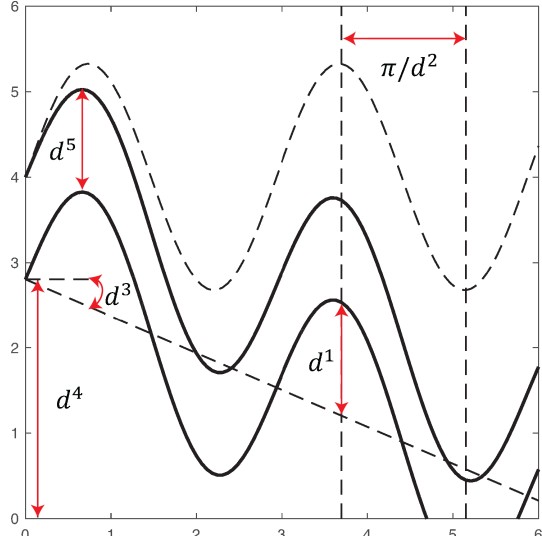

**Figure 1.** Geometrical configuration of channel flow: amplitude $d^1$, frequency $d^2$, angle $d^3$, initial point $d^4$ and width $d^5$.

main $D = [0, 6] \times [0, 6]$, which is given by

$$-\nabla \cdot \left[ k(x, y) \nabla P(x, y) \right] = f(x, y), (x, y) \in D, \qquad (16)$$

where $\nabla = (\partial/\partial x \; \partial/\partial y)'$, $\cdot$ the dot product, $P(x, y)$ the pressure, $k(x, y)$ the permeability, $f(x, y)$ the source term which accounts for groundwater recharge and $(x, y)$ the horizontal dimensions. The boundary conditions are

$$P(x, 0) = 100, \frac{\partial P}{\partial x}(6, y) = 0,$$
$$-k(0, y)\frac{\partial P}{\partial x}(0, y) = 500, \frac{\partial P}{\partial y}(x, 6) = 0, \qquad (17)$$

where $\partial D$ is the boundary of domain $D$. The source term is

$$f(x, y) = \begin{cases} 0 & \text{if } 0 < y \le 4, \\ 137 & \text{if } 4 < y < 5, \\ 274 & \text{if } 5 < y \le 6. \end{cases}$$

We implement a cell-centred finite-difference method and a linear algebra solver (backslash operator in MATLAB) to solve the forward model (Eqs. 16–17) on an $N \times N$ grid.

We note that a single-phase Darcy flow model, though not a steady-state model, is widely used to model the flow in a subsurface aquifer and to infer uncertain permeability using data assimilation. For example, Zovi et al. (2017) used an EKI to infer permeability of an existing aquifer located in north-east Italy. The area of this aquifer is $2.7\,\text{km}^2$ and exhibits several channels, such as the one depicted in Fig. 1. There, a size of a computational cell ranged from 2 m (near wells) to 20 m away from the wells.

## 3.1 Parameterisation of permeability

We consider the following two parameterisations of the permeability function $k(x, y)$:

F1: log permeability over the entire domain $D$, $u(x, y) = \log k(x, y)$;

F2: permeability over domain $D$ that has a channel, $k(x, y) = k^1(x, y)\delta_{D_c}(x, y) + k^2(x, y)\delta_{D \setminus D_c}(x, y)$ as by Iglesias et al. (2014).

Here $D_c$ denotes a channel, $\delta$ is the Dirac function and $k^1 = \exp(u^1(x, y))$ and $k^2 = \exp(u^2(x, y))$ denote permeabilities inside and outside the channel. The geometry of the channel is parameterised by five parameters $\{d^i\}_{i=1}^5$: amplitude, frequency, angle, initial point and width, correspondingly. The lower boundary of the channel is given by $y = d^1 \sin(d^2 x/6) + \tan(d^3)x + d^4$. The upper boundary of the channel is given by $y + d^5$. These parameters are depicted in Fig. 1.

We assume that the log permeability for both F1 and F2 is drawn from a Gaussian distribution $\mu_0 = \mathcal{N}(m, C)$ with mean $m$ and covariance $C$. We define $C$ via a correlation function given by the Whittle–Matern correlation function defined by Matérn (1986):

$$c(x, y) = \frac{1}{\gamma(1)} \frac{\|x - y\|}{\upsilon} \Upsilon_1 \left( \frac{\|x - y\|}{\upsilon} \right),$$

where $\gamma$ is the gamma function, $\upsilon = 0.5$ is the characteristic length scale and $\Upsilon_1$ is the modified Bessel function of the second kind of order 1.

With $\boldsymbol{\lambda}$ and $\mathbf{V}$ we denote eigenvalues and eigenfunctions of the corresponding covariance matrix $\mathbf{C}$, respectively. Then, following a Karhunen–Loève (KL) expansion, log permeability is

$$\log(k^l) = \log(m) + \sum_{\ell=1}^{N^2} \sqrt{\lambda^\ell} V^{\ell l} u^\ell \text{ for } l = 1, \ldots, N^2,$$

where $u^\ell$ is i.i.d. from $\mathcal{N}(0, 1)$ for $\ell = 1, \ldots, N^2$.

For F1, the prior for log permeability is a Gaussian distribution with mean 5. The grid dimension is $N = 70$, and thus the uncertain parameter $\boldsymbol{u} = \{u^\ell\}_{\ell=1}^{N^2}$ has dimension 4900.

For F2, we assume geometrical parameters $\boldsymbol{d} = \{d^i\}_{i=1}^5$ are drawn from uniform priors, namely $d^1 \sim U[0.3, 2.1]$, $d^2 \sim U[\pi/2, 6\pi]$, $d^3 \sim U[-\pi/2, \pi/2]$, $d^4 \sim U[0, 6]$, $d^5 \sim U[0.12, 4.2]$. Furthermore, we assume independence between geometric parameters and log permeability. The prior for log permeability is a Gaussian distribution with mean 15 outside the channel and with mean 100 inside the channel. The grid dimension is $N = 50$. Log permeability inside channel $\boldsymbol{u}^1 = \{u^{1,\ell}\}_{\ell=1}^{N^2}$ and log permeability outside channel $\boldsymbol{u}^2 = \{u^{2,\ell}\}_{\ell=1}^{N^2}$ are defined over the entire domain

$50 \times 50$. Therefore, for F2 inference the uncertain parameter $\boldsymbol{u} = \{\boldsymbol{d}, \boldsymbol{u}^1, \boldsymbol{u}^2\}$ has dimension 5005. Moreover, for F2 we use the Metropolis-within-Gibbs methodology following Iglesias et al. (2014) to separate geometrical parameters and log permeability parameters within the mutation step, since it allows the structure of the prior to be better exploited.

## 3.2 Observations

Both the true permeability and an initial ensemble are drawn from the same prior distribution as the prior includes knowledge about geological properties. However, an initial guess is computed on a coarse grid, and the true solution is computed on a fine grid that has twice the resolution of the coarse grid. The synthetic observations of pressure are obtained by

$$\boldsymbol{y}_{\mathrm{obs}} = \boldsymbol{L}(\boldsymbol{P}^{\mathrm{true}}) + \boldsymbol{\eta}.$$

An element of $\boldsymbol{L}(\boldsymbol{P}^{\mathrm{true}})$ is a linear functional of pressure, namely

$$L^j(\boldsymbol{P}^{\mathrm{true}}) = \frac{1}{2\pi\sigma^2} \sum_{i=1}^{N_f} \exp\left(-\frac{\|\boldsymbol{X}^i - \boldsymbol{h}^j\|^2}{2\sigma^2}\right)(P^{\mathrm{true}})^j \Delta x^2$$

$$\text{for } j = 1, \ldots, \kappa.$$

Here $\sigma = 0.01$, $\Delta x^2$ is the size of a grid cell $\boldsymbol{X}^i = (X^i, Y^i)$, $N_f$ is the resolution of a fine grid, $\boldsymbol{h}^j$ is the location of the observation and $\kappa$ is the number of observations. This form of the observation functional and the parameterisation F1 and F2 guarantee the continuity of the forward map from the uncertain parameters to the observations and thus the existence of the posterior distribution, as shown by Iglesias et al. (2014). The observation noise $\boldsymbol{\eta}$ is drawn from a normal distribution with zero mean and known covariance matrix $\mathbf{R}$. We choose the observation noise to be $2\%$ of L2-norm of the true pressure. With such a small noise the likelihood is a peaked distribution. Therefore, a non-iterative data assimilation approach requires a computationally unfeasible number of ensemble members to sample the posterior.

To save computational costs, we choose an ESS threshold $M_{\mathrm{thresh}} = M/3$ for tempering and the length of the Markov chain $\tau_{\max} = 20$ for mutation.

## 3.3 Metrics

We conduct numerical experiments with ensemble sizes $M = 100$ and $M = 500$ and 20 simulations with different initial ensemble realisations to check the robustness of results. We analyse the method's performance with respect to a pcn-MCMC solution, from here on referred to as the reference. An MCMC solution was obtained by combining 50 independent chains each of length $10^6$, $10^5$ burn-in period and $10^3$ thinning. For log permeability, we compute the root mean square error (RMSE) of the mean

$$\mathrm{RMSE} = \sqrt{(\bar{\boldsymbol{u}} - \bar{\boldsymbol{u}}^{\mathrm{ref}})'(\bar{\boldsymbol{u}} - \bar{\boldsymbol{u}}^{\mathrm{ref}})}, \text{ where } \bar{\boldsymbol{u}} = \frac{1}{M}\sum_{i=1}^{M} \boldsymbol{u}_i, \quad (18)$$

and $\boldsymbol{u}^{\mathrm{ref}}$ is the reference solution.

For geometrical parameters $\boldsymbol{d}$, we compute the Kullback–Leibler divergence

$$\mathrm{D}_{\mathrm{KL}}^i(p^{\mathrm{ref}} \parallel p) = \sum_{j=1}^{M_b} p^{\mathrm{ref}}(\mathrm{d}_j^i) \log \frac{p^{\mathrm{ref}}(\mathrm{d}_j^i)}{p(\mathrm{d}_j^i)}, \quad (19)$$

where $p^{\mathrm{ref}}(\mathrm{d}^i)$ is the reference posterior, $p(\mathrm{d}^i)$ is approximated by the weights and $M_b = M/10$ is a chosen number of bins.

## 3.4 Application to F1 inference

For F1, we perform numerical experiments using 36 uniformly distributed observations, which are displayed in circles in Fig. 3a. We plot a box plot of the RMSE given by Eq. (18) over 20 independent simulations in Fig. 2a using Sinkhorn approximation and in Fig. 2b using optimal transport. The horizontal axis is for the hybrid parameter $\beta$, whose value 0 corresponds to the EKI and 1 to an adaptive SMC method with either a Sinkhorn approximation (TESPF) or optimal transport (TETPF). Ensemble size $M = 100$ is shown in red and $M = 500$ in green. First, we observe that at a small ensemble size $M = 100$ and a large $\beta$ (namely $\beta \geq 0.6$), TESPF outperforms the TETPF as the RMSE is lower. Since Sinkhorn approximation is a regularisation of an optimal transport solution, the TESPF provides a smoother solution than the TETPF that can be seen in Fig. 3c and 3f, respectively, where we plot mean log permeability. Next, we see in Fig. 2 that the hybrid approach decreases the RMSE compared to TET(S)PF: the smaller the $\beta$ value, the smaller the median of the RMSE. The EKI gives the smallest error due to the Gaussian parameterisation of permeability. The advantage of the hybrid approach is most pronounced at a large ensemble size of $M = 500$ and optimal transport resampling. Furthermore, we note a discrepancy between the $M = 100$ and the $M = 500$ experiments at $\beta = 0$, i.e. for the pure EKI. This is related to the curse of dimensionality. It appears that the ensemble size $M = 100$ is too small to estimate an uncertain parameter of the dimension $10^3$ using 36 accurate observations. However, at the ensemble size $M = 500$ the EKI alone ($\beta = 0$) gives an excellent performance compared to any combination ($\beta > 0$).

We plot mean log permeability at ensemble size $M = 100$ and the smallest RMSE over 20 simulations in Fig. 3b–f and of the reference in Fig. 3a. We see that the EKI and the TETPF(0.2) estimate not only large-scale features but also small-scale features (e.g. negative mean in the top right corner) unlike the TET(S)PF and TESPF(0.2) well.

## 3.5 Application to F2 inference

For F2, we perform numerical experiments using nine uniformly distributed observations. which are displayed in circles in Fig. 9a. First, we display results obtained using

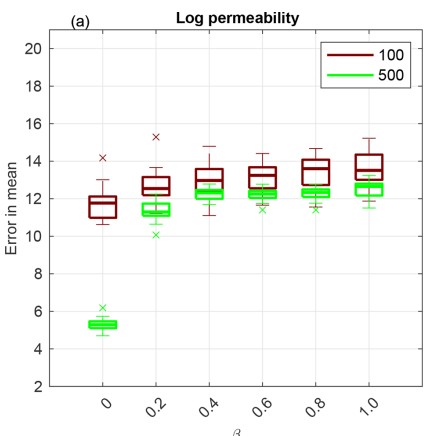
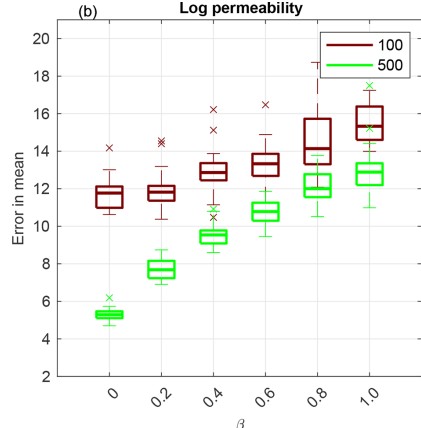

**Figure 2.** Application to F1 parameterisation: using Sinkhorn approximation **(a)** and optimal transport resampling **(b)**. Box plot over 20 independent simulations of the RMSE of mean log permeability. The horizontal axis is for the hybrid parameter, where $\beta = 0$ corresponds to the EKI and $\beta = 1$ to TET(S)PF. The ensemble size $M = 100$ is shown in red and $M = 500$ in green. The central mark is the median, the edges of the box are the 25th and 75th percentiles, whiskers extend to the most extreme data points and crosses are outliers.

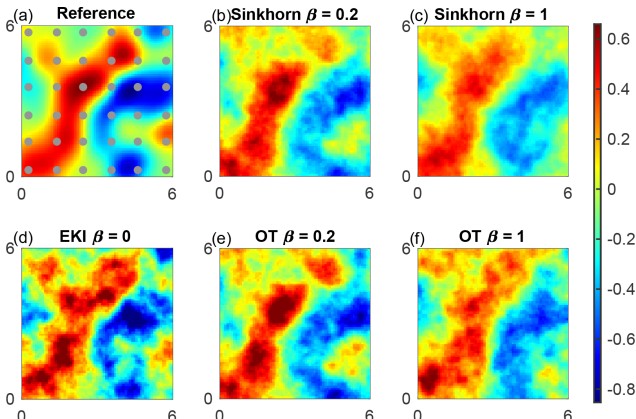

**Figure 3.** Mean log permeability for F1 inference for the lowest error at ensemble size $M = 100$. Observation locations are shown by circles. Reference **(a)**, TESPF(0.2) **(b)**, TESPF **(c)**, EKI **(d)**, TETPF(0.2) **(e)** and TETPF **(f)**.

Sinkhorn approximation. In Fig. 4, we plot a box plot over 20 independent runs of KL divergence given by Eq. (19) for amplitude (a), frequency (b), angle (c), initial point (d) and width (e) that define the channel. We see that the EKI outperforms any TESPF(·) including the TESPF for amplitude (a) and width (e). This is due to Gaussian-like posteriors of these two geometrical parameters displayed in Fig. 6c and 6o, respectively. Due to Gaussian-like posteriors the hybrid approach decreases the RMSE compared to the TESPF: the smaller the $\beta$ value, the smaller the median of the RMSE.

For frequency, angle and initial point, whose KL divergence is displayed in Fig. 4b, c and d, respectively, the behaviour of adaptive SMC is non-linear in terms of $\beta$. This is due to non Gaussian-like posteriors of these three geometrical parameters shown in Fig. 6f, i and l, respectively. Due

to non Gaussian-like posteriors, the hybrid approach gives an advantage over both the TESPF and the EKI – there is a $\beta \neq 0$ for which the KL divergence is lowest, although it is inconsistent between geometrical parameters.

When comparing the TESPF(·) to the TETPF(·), we observe the same type of behaviour in terms of $\beta$: linear for amplitude and width, whose KL divergence is displayed in Fig. 5a and e, respectively, and non-linear for frequency, angle and initial point, whose KL divergence is displayed in Fig. 5b, c and d, respectively. However, the KL divergence is smaller when optimal transport resampling is used instead of Sinkhorn approximation.

In Fig. 6, we plot the posteriors of geometrical parameters: amplitude (a–c), frequency (d–f), angle (g–i), initial point (j–l) and width (m–o); on the left the TESPF(0.2) is shown, in the middle the TETPF(0.2) and on the right the EKI. In black is the reference, in red 20 simulations of ensemble size $M = 100$ and in green 20 simulations of ensemble size $M = 500$. The true parameters are shown by black crosses. We see that as the ensemble size increases, posteriors approximated by TET(S)PF converge to the reference posterior unlike the EKI.

Now we investigate adaptive SMC performance for permeability estimation. First, we display results obtained using Sinkhorn approximation. The box plot shows over 20 independent simulations of the RMSE given by Eq. (18) for log permeability outside the channel in Fig. 7a and inside the channel in Fig. 7b. Even though log permeability is Gaussian-distributed, for a small ensemble size $M = 100$, there is a $\beta \neq 0$ that gives the lowest RMSE both outside and inside the channel. As the ensemble size increases, the methods' performance becomes equivalent.

Next, we compare the TESPF(·) to the TETPF(·) for log permeability estimation outside and inside the channel, whose RMSE is displayed in Fig. 8a and b, respectively.

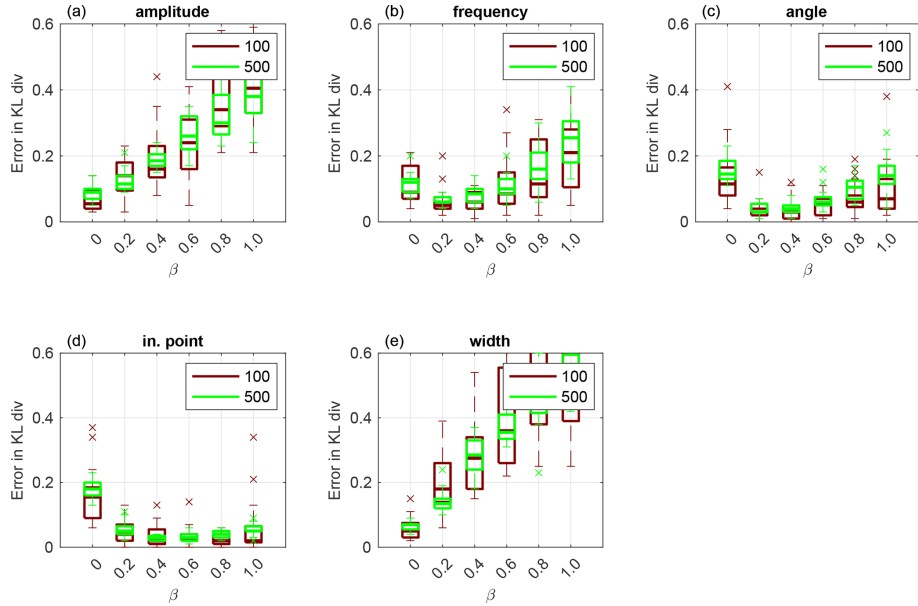

**Figure 4.** Application to F2 parameterisation using Sinkhorn approximation. Box plot over 20 independent simulations of KL divergence for geometrical parameters: amplitude **(a)**, frequency **(b)**, angle **(c)**, initial point **(d)** and width **(e)**. The horizontal axis is for the hybrid parameter, where $\beta = 0$ corresponds to the EKI and $\beta = 1$ to TET(S)PF. Ensemble size $M = 100$ is shown in red and $M = 500$ in green. The central mark is the median, the edges of the box are the 25th and 75th percentiles, whiskers extend to the most extreme data points and crosses are outliers.

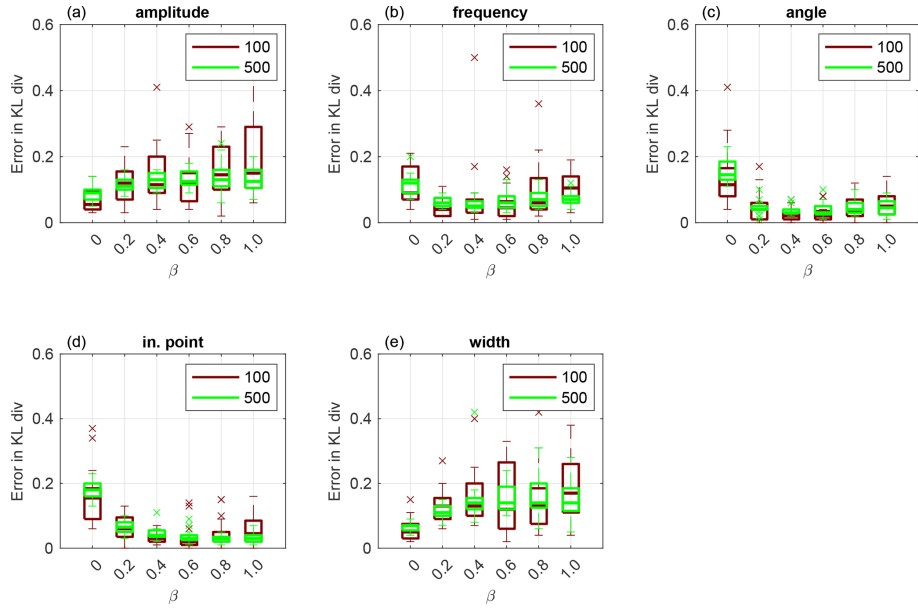

**Figure 5.** The same as Fig. 4 but using optimal transport resampling.

We observe the same type of behaviour in terms of $\beta$: non-linear for a small ensemble size $M = 100$ and equivalent for a larger ensemble size $M = 500$. Furthermore, at a small ensemble size $M = 100$, the TESPF outperforms the TETPF, which was also the case for F1 parameterisation (Sect. 3.4).

In Fig. 9, we show the mean field of permeability over the channelised domain for reference for the lowest error at

ensemble size $M = 100$ for the TESPF(0.2) (b), TESPF (c), EKI (d), TETPF(0.2) (e) and TETPF (f). We plot mean log permeability over the channelised domain at ensemble size $M = 100$ and the smallest RMSE over 20 simulations in Fig. 9b–f and for the reference in Fig. 9a. We see that TESPF(0.2) does an excellent job at such a small ensemble

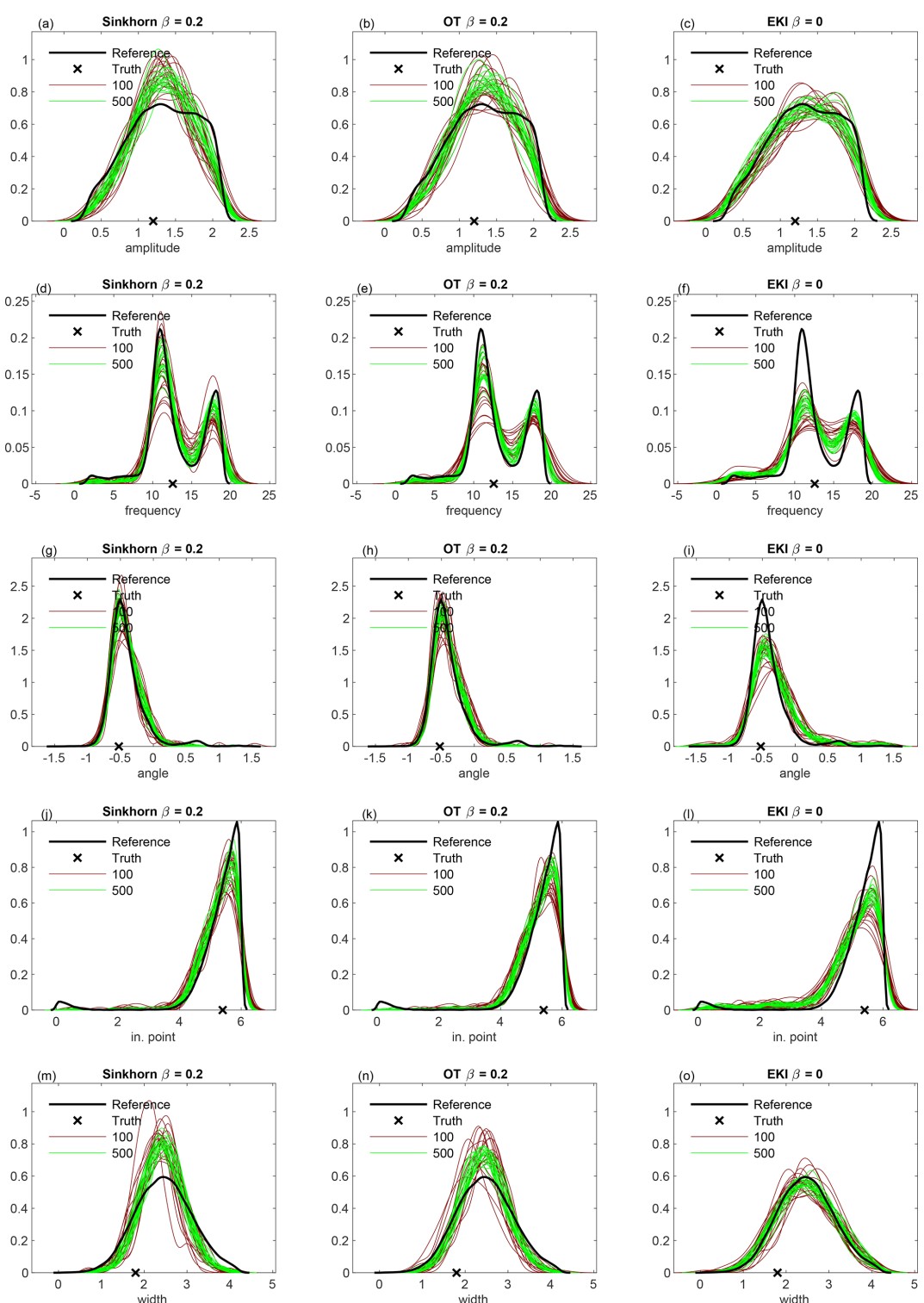

**Figure 6.** Posteriors of geometrical parameters for F2 inference: amplitude **(a–c)**, frequency **(d–f)**, angle **(g–i)**, initial point **(j–l)** and width **(m–o)**. On the left is the TESPF(0.2), in the middle the TETPF(0.2) and on the right the EKI. In black is the reference, in red 20 simulations of ensemble size $M = 100$ and in green 20 simulations of ensemble size $M = 500$. The true parameters are shown by black crosses.

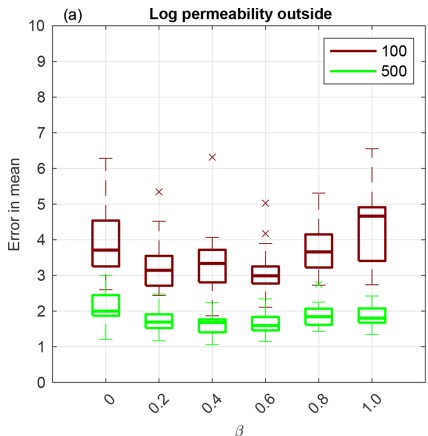 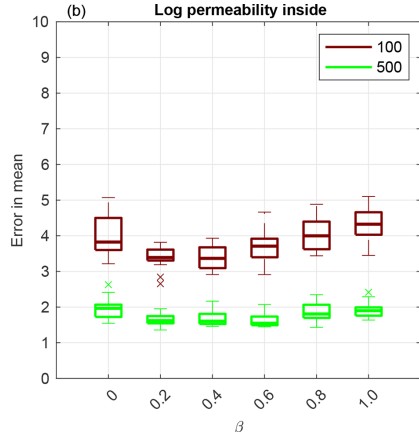

**Figure 7.** Application to F2 parameterisation using Sinkhorn approximation. Box plot over 20 independent simulations of RMSE of mean log permeability outside the channel (**a**) and inside the channel (**b**). The horizontal axis is for the hybrid parameter, where $\beta = 0$ corresponds to the EKI and $\beta = 1$ to TET(S)PF. The ensemble size $M = 100$ is shown in red and $M = 500$ in green. The central mark is the median, the edges of the box are the 25th and 75th percentiles, whiskers extend to the most extreme data points and crosses are outliers.

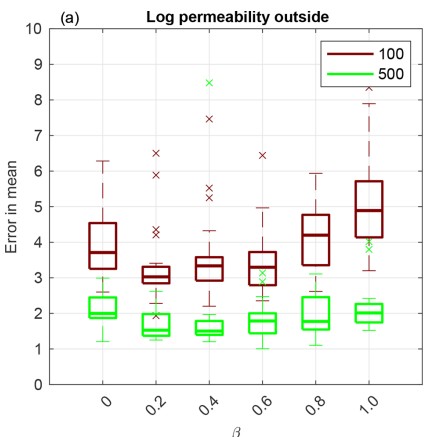 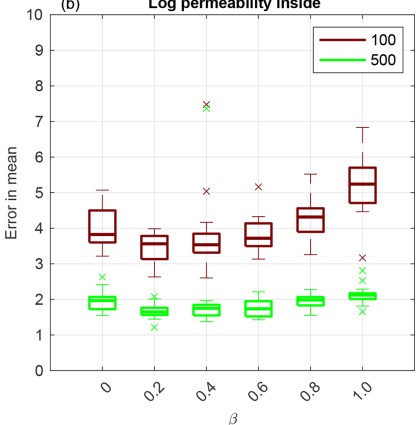

**Figure 8.** The same as Fig. 7 but using optimal transport resampling.

size by estimating log permeability outside and inside the channel well, as well as parameters of the channel itself.

## 4 Conclusions

A Sinkhorn adaptation, namely the TESPF, of the previously proposed TETPF has been introduced and numerically investigated for a parameter estimation problem. The TESPF has similar accuracy results to the TETPF (see Figs. 6, 7 and 8), while it can have considerably smaller computational complexity. Specifically, the TESPF has a complexity $\mathcal{O}[T(M\mathcal{C}+M^2\mathcal{C}(\alpha)+\tau_{\max}M\mathcal{C})]$ and the TETPF $\mathcal{O}[T(M\mathcal{C}+M^3\log M+\tau_{\max}M\mathcal{C})]$ (for a complete overview, see Table B1). In particular, the TESPF outperforms the EKI for non-Gaussian distributed parameters (e.g. initial point and angle in F2). This makes the proposed method a promising option for the high dimensional non-linear problems one

is typically faced with in reservoir engineering. Further, to counterbalance potential robustness problems of the TETPF and its Sinkhorn adaptation, a hybrid between the EKI and the TET(S)PF is proposed and studied by means of the two configurations of the steady-state single-phase Darcy flow model. The combination of the two adaptive SMC methods with complementing properties, i.e. $\beta \in (0,1)$, is superior to the individual adaptive SMC method, i.e. $\beta = 0$ or 1, for all non-Gaussian distributed parameters and performs better than the pure TETPF and the TESPF for Gaussian distributed parameters in F1. This suggests a hybrid approach has a great potential to obtain robust and highly accurate approximate solutions of non-linear high dimensional Bayesian inference problems. Note that we have considered a synthetic case, where the truth is available, and thus chose $\beta$ in terms of accuracy of an estimate. However, in a realistic application, the truth is not provided. In the context of state estimation with an underlying dynamical system, it has been suggested

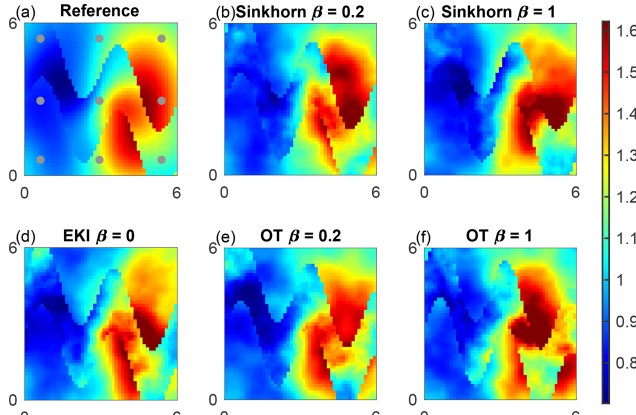

**Figure 9.** Mean log permeability for F2 inference for the lowest error at ensemble size $M = 100$. Observation locations are shown by circles. Reference **(a)**, TESPF(0.2) **(b)**, TESPF **(c)**, EKI **(d)**, TETPF(0.2) **(e)** and TETPF **(f)**.

to adaptively change the hybrid parameter with respect to the effective sample size. As the tempering scheme is already changed according to the effective sample size, this ansatz would require the interplay between the two tuning variables to be defined. An ad hoc choice for $\beta$ could be 0.2 or 0.3. This is motivated by the fact that the particle filter is too unstable in high dimensions, and it is therefore sensible to use a tuning parameter prioritising the EKI. The ad hoc choice is supported by the numerical results in Sect. 3 and in Acevedo et al. (2017) and de Wiljes et al. (2020) in the context of state estimation.

## Appendix A: Pseudocode

---

**Algorithm 1** Sample mutation

---

**Require:** $\theta \in (0,1)$ and an integer $\tau_{\max}$
    **for** $i = 1, \ldots, M$ **do**
        Initialize $v_i(0) = \tilde{u}_{t,i}$
        **while** $\tau \leq \tau_{\max}$ **do**
            Propose $v_i^{\text{prop}}$ using Eq. (5) for Gaussian probability or Eq. (6) for uniform probability
            Set $v_i(\tau + 1) = v_i^{\text{prop}}$ with probability Eq. (7) and set $v_i(\tau + 1) = \tilde{u}_{t,i}$ with probability Eq. (8)
            $\tau \leftarrow \tau + 1$
        **end while**
        Set $u_{t,i} = v_i(\tau_{\max})$
    **end for**

---

**Algorithm 2** Resampling based on optimal transport

---

**Require:** $\{u_{t-1,i}\}_{i=1}^{M}$ and $w_{t-1} = \{w_{t-1,1}, \ldots, w_{t-1,M}\}$
    Compute $\mathbf{Z}$ with $z_{ij} = \|u_{t-1,i} - u_{t-1,j}\|^2$
    Supply $\mathbf{Z}$ and $w_{t-1}$ to the FastEMD algorithm of Pele & Werman with the output being the coupling $\mathbf{S}$
    Compute new samples $\{\tilde{u}_{t,i}\}_{i=1}^{M}$ from Eq. (11)

---

**Algorithm 3** Sinkhorn iteration for optimal transport problem with entropic regularisation

---

**Require:** regularisation parameter $\alpha$, $\{u_{t-1,i}\}_{i=1}^{M}$ and $w_{t-1} = \{w_{t-1,1}, \ldots, w_{t-1,M}\}$
    Compute $\mathbf{Z}$ with $z_{ij} = \|u_{t-1,i} - u_{t-1,j}\|^2$
    Normalise $\mathbf{Z}$ with respect to its maximum entry
    **while** $\varepsilon \geq 1.0e - 8$ **do**
        $b = w_{t-1}./[\exp(-\alpha \mathbf{Z})a]$
        $a = \left(\frac{1}{M} I_M / M\right)./[\exp(-\alpha \mathbf{Z})b]$
        $\mathbf{S} = \mathbf{diag}(b)\exp(-\alpha \mathbf{Z})\mathbf{diag}(a)$
        $\hat{w} = \mathbf{S}I_M$
        $\varepsilon = \|\hat{w} - w_{t-1}\|$
    **end while**
    **return** $\mathbf{S}^* = \mathbf{S}$

---

---

**Algorithm 4** Adaptive SMC: TET(S)PF

---

**Require:** an initial ensemble $\{u_{0,i}\}_{i=1}^{M} \sim \mu_0$, $\theta \in (0,1)$ and integers $\tau_{max}$ and $1 < M_{thresh} < M$

Set $\phi_0 = 0$

**while** $\phi_t \leq 1$ **do**

$\quad t \to t+1$

$\quad$ Compute the likelihood $g(u_{t-1,i}; y_{obs})$ from Eq. (2) (for $i = 1, \ldots, M$)

$\quad$ Compute the tempering parameter $\phi_t$:

$\quad$ **if** $\min_{\phi \in (\phi_{t-1},1)} \mathrm{ESS}_t(\phi) > M_{thresh}$ **then**

$\quad\quad$ set $\phi_t = 1$

$\quad$ **else**

$\quad\quad$ compute $\phi_t$ such that $\mathrm{ESS}_t(\phi) \approx M_{thresh}$ using a bisection algorithm on $(\phi_{t-1}, 1]$

$\quad$ **end if**

$\quad$ Compute weights $w_{t-1} = \{w_{t-1,1}, \ldots, w_{t-1,M}\}$ from Eq. (3)

$\quad$ Create new samples $\{\tilde{u}_{t,i}\}_{i=1}^{M}$ using optimal (Sinkhorn) resampling via Algorithm 2(3)

$\quad$ Compute $\{u_{t,i}\}_{i=1}^{M}$ using sample mutation via Algorithm 1

**end while**

---

**Algorithm 5** EKI

---

**Require:** an initial ensemble $\{u_{0,i}\}_{i=1}^{M} \sim \mu_0$, $\theta \in (0,1)$ and integers $\tau_{max}$ and $1 < M_{thresh} < M$

Set $\phi_0 = 0$

**while** $\phi_t \leq 1$ **do**

$\quad t \to t+1$

$\quad$ Compute the likelihood $g(u_{t-1,i}; y_{obs})$ from Eq. (2) (for $i = 1, \ldots, M$)

$\quad$ Compute the tempering parameter $\phi_t$:

$\quad$ **if** $\min_{\phi \in (\phi_{t-1},1)} \mathrm{ESS}_t(\phi) > M_{thresh}$ **then**

$\quad\quad$ set $\phi_t = 1$

$\quad$ **else**

$\quad\quad$ compute $\phi_t$ such that $\mathrm{ESS}_t(\phi) \approx M_{thresh}$ using a bisection algorithm on $(\phi_{t-1}, 1]$

$\quad$ **end if**

$\quad$ Create new samples $\{\tilde{u}_{t,i}\}_{i=1}^{M}$ using Eq. (13)

$\quad$ Compute $\{u_{t,i}\}_{i=1}^{M}$ using sample mutation via Algorithm 1

**end while**

---

---

**Algorithm 6** Hybrid EKI-TET(S)PF

---

**Require:** initial initial ensemble $\{u_{0,i}\}_{i=1}^{M} \sim \mu_0$, $\theta \in (0,1)$, hybrid parameter $\beta$ and integers $\tau_{\max}$ and $1 < M_{\text{thresh}} < M$

    Set $\phi_0 = 0$

    **while** $\phi_t \leq 1$ **do**

        $t \rightarrow t + 1$

        Compute the likelihood $g_1(u_{t-1,i}; y_{\text{obs}})$ from Eq. (14) (for $i = 1, \ldots, M$)

        Set $g(u_{t-1,i}; y_{\text{obs}}) = g_1(u_{t-1,i}; y_{\text{obs}})$ (for $i = 1, \ldots, M$)

        Compute the tempering parameter $\phi_t$:

        **if** $\min_{\phi \in (\phi_{t-1}, 1)} \text{ESS}_t(\phi) > M_{\text{thresh}}$ **then**

            set $\phi_t = 1$

        **else**

            compute $\phi_t$ such that $\text{ESS}_t(\phi) \approx M_{\text{thresh}}$ using a bisection algorithm on $(\phi_{t-1}, 1]$

        **end if**

        Create new samples $\{\tilde{u}_{t,i}^{\beta}\}_{i=1}^{M}$ using Eq. (13)

        Compute the likelihood $g_2(\tilde{u}_{t,i}^{\beta}; y_{\text{obs}})$ from Eq. (15) (for $i = 1, \ldots, M$)

        Set $g(u_{t-1,i}; y_{\text{obs}}) = g_2(\tilde{u}_{t,i}^{\beta}; y_{\text{obs}})$ (for $i = 1, \ldots, M$)

        Compute weights $w_{t-1} = \{w_{t-1,1}, \ldots, w_{t-1,M}\}$ from Eq. (3)

        Create new samples $\{\tilde{u}_{t,i}\}_{i=1}^{M}$ using optimal (Sinkhorn) resampling via Algorithm 2(3)

        Compute $\{u_{t,i}\}_{i=1}^{M}$ using sample mutation via Algorithm 1

    **end while**

---

## Appendix B: Computational complexity

**Table B1.** The table provides an overview of the computational complexity of all the algorithms considered in the paper.

| Algorithm | Complexity |
|---|---|
| TETPF | $\mathcal{O}[T(M\mathcal{C} + M^3 \log M + \tau_{\max} M\mathcal{C})]$ |
| TESPF | $\mathcal{O}[T(M\mathcal{C} + M^2 C(\alpha) + \tau_{\max} M\mathcal{C})]$ |
| EKI | $\mathcal{O}[T(M\mathcal{C} + \kappa^2 n + \tau_{\max} M\mathcal{C})]$ |
| Hybrid EKI–TETPF | $\mathcal{O}[T(M\mathcal{C} + \kappa^2 n + M\mathcal{C} + M^3 \log M + \tau_{\max} M\mathcal{C})]$ |
| Hybrid EKI–TESPF | $\mathcal{O}[T(M\mathcal{C} + \kappa^2 n + M\mathcal{C} + M^2 C(\alpha) + \tau_{\max} M\mathcal{C})]$ |
| Forward model $G$ | $\mathcal{O}(M\mathcal{C})$ |
| pcn-MCMC mutation | $\mathcal{O}(\tau_{\max} M\mathcal{C})$ |
| FastEMD | $\mathcal{O}(M^3 \log M)$ |
| Sinkhorn approximation | $\mathcal{O}(M^2 C(\alpha))$ |

*Code and data availability.* Data and MATLAB codes for generating the plots are available in Ruchi et al. (2021).

*Author contributions.* SR, SD and JdW designed the research. SD ran the numerical experiments. SR, SD and JdW analysed the results and wrote the paper.

*Competing interests.* The authors declare that they have no conflict of interest.

*Acknowledgements.* The research of Jana de Wiljes and Sangeetika Ruchi have been partially funded by the Deutsche Forschungsgemeinschaft (DFG) SFB 1294/1 – 318763901. Further, Jana de Wiljes has been supported by Simons CRM Scholar-in-Residence Program and ERC Advanced Grant ACRCC (grant no. 339390). Sangeetika Ruchi has been supported by the research programme Shell-NWO/FOM Computational Sciences for Energy Research (CSER), project no. 14CSER007, which is partly financed by the Netherlands Organization for Scientific Research (NWO).

*Financial support.* This research has been supported by the Deutsche Forschungsgemeinschaft (grant no. SFB1294/1 – 318763901), the European Research Council (Advanced Grant ACRCC (grant no. 375 339390)), the Nederlandse Organisatie voor Wetenschappelijk Onderzoek, Stichting voor de Technische Wetenschappen (grant no. Shell-NWO/FOM Computational Sciences for Energy Research (CSER), project no. 14CSER007) and the Simons Foundation (CRM Scholar-in-Residence Program grant).

*Review statement.* This paper was edited by Olivier Talagrand and reviewed by Femke Vossepoel and Marc Bocquet.

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

TS9 Please confirm volume and page range.