# Peer review of "Application of ensemble transform data assimilation methods for parameter estimation in nonlinear problems"

_Nonlinear Processes in Geophysics, 2020_

## Referee Comment (RC1) · Femke Vossepoel (Referee) · 28 Jul 2020

Review of 'Application of ensemble transform data assimilation methods for parameter estimation in nonlinear problems', by Sangeetika Ruchi, Svetlana Bubinkina and Jana de Wiljes.

**Generall comments**

This manuscript describes an overview and comparison of a number of data-assimilation methods for parameter estimation in nonlinear problems. It describes the tempered ensemble transform particle filter as an alternative to Ensemble Kalman Inversion. To reduce computational costs, the authors introduce an entropy-inspired regularisation factor to underlying transport problem. The application of a Sinkhornfixpoint iteration reduces the computational costs considerably. This is a new addition to existing particle filters. The manuscript further discusses two different hybrid approaches that apply a Kalman-inversion proposal step in the particle filter. The menchmark of these methods against a Markov-Chain Monte Carlo approach as well as the assessment of their computational costs is a valuable and significant result. The description of the computational complexity help to quantify the efficiency of the methods. An example of Darcy flow through a synthetic reservoir (aquifer) illustrates the quality of the parameter estimate and the general performance of the methods.

This is an interesting manuscript that deserves publication. The comparison of the different methods, and the clear explanation of the underlying mathematics will help data-assimilation practitioners to make a balanced choice between several data-assimilation methods. By presenting original solutions, the manuscript inspires those who develop data-assimilation methods to further refinement of methods or innovative approaches. The language is clear and concise, but not all symbols used are explained, and some terms could be clarified further. The paper puts the obtained results into context, but could relate the results more to the field of the proposed application and include relevant references of that field. The text reads well and the figures are of good quality.

I hesitate between 'accepted with minor revision' and 'accepted with major revision'. The manuscript would be publishable with only minor text adaptations, but I feel it could have a much larger impact if the authors would change the manuscript more substantially. There are two main items that I would encourage the authors to address. In addition to this, I have a list of minor issues, mostly textual.

**Specific comments**

Terminology and description of example: As this paper could be of particular use to practitioners in the reservoir-engineering domain, I would encourage the authors to make the text more accessible to those. This could be done by changing or clarifying the use of certain terms and adding key references to explain the methods. For example, in reservoir engineering, the term Ensemble Kalman Filter is more commonly used than the term Ensemble Kalman Inversion; adding a number of key publications on this method and derived methods would help to set the scene and provide the reader with further background information. Also, those using data assimilation in practical applications will be

interested in the actual values of the properties, and less likely to work on dimensionless problems. Relating the symbols to physical quantities would make this manuscript more accessible and relevant to them.

Presentation of the methodologies: The mathematical rigour and expertise of the authors would allow them to not only compare the performance of the methods in an empirical sense, but also place them in the overall framework of data-assimilation methods for parameter estimation. The manner in which the hybrid EKI-TETPF method is presented, is presented as a particle filter with several 'fixes' (namely a) tempering, b) a Sinkhorn approximation, and c) an EKI proposal). Can the authors think of a way to present the methods from a holistic viewpoint, making clear that these 'fixes' are essential ingredients of the methods in order to perform a consistent and also effective parameter estimation? The abstract reads "Gaussian approximations [....] often produce astonishingly accurate estimations despite the inherently wrong underlying assumptions": Can you discuss more explitly how the assumption of Gaussianity affects the outcome, perhaps by illustrating how non-Gaussian the distributions really are, or how the different methods deal with non-Gaussianity and/or non-linearity?

**Technical corrections (language, minor items)**

- Please pay attention to the use of hyphens in compound modifiers. For example, the title could read 'Application of ensemble-transform data-assimilation methods for parameter estimation in nonlinear problems'. Other places where this would help: 'high-dimensional problems', 'entropy-inspired', 'highly-correlated samples',  'an easy-to-sample form', etc.
- The term 'ensemble Kalman inversion' is used to a method that is known by many as 'ensemble Kalman filtering'. I suggest to clarify that EKI is used as equivalent to the ensemble Kalman filter. Page 2, line 38 and/or in the paragraph startig on p.8, line 201: suggest to add one of the key references for ensemble Kalman inversion or ensemble Kalman filtering, so readers can find out more about the method.
- Page 1, line 3: abstract: 'inherently wrong': the Gaussian assumptions are not always wrong, so suggest to reformulate: 'depsite the simplifying assumptions' or something along these lines. Alternatively, demonstrate in the manuscript that these assumptions are actually wrong.
- Page 2, line 55: "the number of required intermediate steps and the efficiency of ETPF still depends on it". What does "it" refer to?
- Page  5, line 118: Crank-Nicholson pcn-MCMC: explain what pcn means here.
- Page  5 line 130: the scalar theta -> the scalar theta in Equation 5
- Page 6, line 152: where the minimum is compute -> where the minimum is computed
- Page 7 line 181 One the other hand -> on the other hand
- Page 8, line 204: estimation of posterior -> estimation of the posterior
- Page 8, line 205-215: make sure to list and clarify all symbols used.
- Page 9, line 232: make clear how to choose beta
- Page 9, line 239: EKI as an more elaborate -> as a more elaborate

- Page 9, line 240: Computational complexity: the estimates of computationa complexity of the various methods is very useful. I suggest to include a table that illustrates the computational complexity of all methods/variations and include a few sentences on this in the 'Conclusions' part.
- Page 9, line 244: the example is dimensionless. Suggest to relate this to an example in which you list a number of typical values. You can then also mention that channels such as shown in Fig 1 can be found in fluvial rock formations that form aquifers or reservoirs.
- Page 9, line 247: please make clear what physical variable (rate, pressure) the source term represents
- Page 9 line 267: on an NxN grid: a potential user would like to know what is the scale, and spatial dimension. Suggest to give the value of N earlier than you do now (on page 10, line 273).
- Page 10, line 255: the choice of P for parameterisation is not very practical, as you are also using this letter for pressure
- Page 11, line 285: please make clear what property is being observed
- Page 12, line 310: we plot box plot -> we plot a box plot; using Sinkhorn approximation -> using a Sinkhorn approximation
- Page 12, line 313: TESPF outperforms: has a lower RMSE? Is smoother? How do you define a good performance?
- Page 12 line 320: estimate well mot -> estimate well not
- Page 13, figure 2: it is good to see the box plots for permeability, I would have liked to also see this for rate (observed state variable)
- Page 14 figure 4: please label the x axis (it is described in the caption but would be good to see in the figure, too).
- Page 14, line 332: these are very interesting results. I would value a discussion on how to find the best beta value in a realistic application of the hybrid method. In a synthetic case, this value can be determined, but how would you deal with this when assimilating real data? This discussion could be added in 'conclusions' (page 17)
- Page 14 line 344: we plot box plot -> we plot a box plot (or 'the box plot shows…')
- Page 15, line 363: the application that you show, would be referred to as a 'reservoir engineering' application, or a 'hydrological' application, not as a 'geophysical application'. (In oil- and gas industry, reservoir engineering is about flow in porous media, while geophysics is about the use of seismic and other geophysical data and propagation of sound waves. In hydrology, permeability is usually replaced by hydraulic conductivity, so by using permeability your example would be more familiar to those working in reservoir engineering.)

---

## Referee Comment (RC2) · Marc Bocquet (Referee) · 29 Aug 2020

**1   General opinion**

This is a nicely written paper. It is very technical and the manuscript requires quite a few mathematical pieces of knowledge to follow the methodology! But, in my opinion, this technicality is justified. This, however, requires the notation and naming to be very clear and homogeneous. Although not too numerous, there are quite a few typos that need correcting. The numerical application is very appealing and enlightening, although not

entirely convincing because of the gap in performance between the ensemble sizes 100 and 500 (for the permeability field, not the parameters of the vein).

Overall, I believe the manuscript only requires a minor revision but that it should be very carefully addressed.

**2 Remarks and suggestions:**

1. Page 1: I believe that the title of the paper is too generic, not specific enough. It could suit dozens of papers already published. I strongly suggest that you revise it. I understand that this is not easy since you use a large collection of methods. Although quick to amend, I believe this point is problematic for the visibility/identification of the paper.

2. Page 1, line 2: "Kalman inversion" is not a widespread terminology, "randomized maximum likelihood" is better known, even beyond the reservoir community. See Oliver et al. (1996) and many references since then.

3. Page 1, line 4: "of the associated statistics.": I am not sure to get what you mean.

4. page 1, line 18, "a just approximation": do you mean a "correct approximation"?

5. page 2, line 28, "The main drawback of MCMC is that this approach is not parallelizable.": You know that there are parallel (multiple tries) versions of MCMCs. It actually seems that you are yourself using multiple parallel MCMCs. So I believe you should mitigate that statement.

6. Page 2, line 41-43: "However for nonlinear problems, Ernst et al. (2015); Evensen (2018) showed that in the large ensemble size limit an EKI approximation is not consistent with the Bayesian approximation.": To the best of my knowledge this

is has been pointed out first by Oliver et al. (1996). The mathematical problem has also been clearly defined by Bardsley et al. (2014), and nicely named 'Randomize-Then-Optimize". There is also a recent discussion on the issue in Liu et al. (2017), p. 2894.

7. Page 2, line 57-58: Yes, but you should at this point mention here that the idea originates from the optimal transport community, and that it is by now widespread.

8. Page 3, line 73: even though obvious, it would be better to mention explicitly that $\mathcal{N}$ is the Gaussian distribution.

9. Page 4, line 114: "Mutation" is applied mathematics Pierre Del Moral's terminology. You could briefly explain what it corresponds to in the geophysics particle filter community (rejuvenation?)

10. Page 4, line 122: "we use random walk" $\longrightarrow$ "we use the following random walk"

11. Page 5, line 135: "where C is computational cost of a forward model F" $\longrightarrow$ "where C is the computational cost of the forward model F"

12. Page 6, line 136: 'is not effected" $\longrightarrow$ "is not affected"

13. Page 6, line 150: "we seak" $\longrightarrow$ "we seek"

14. Page 6, line 160, Eq.(10): What is the definition of the norm of the random variables used in this equation?

15. Page 7, line 181: "One the other hand" $\longrightarrow$ "On the other hand"

16. Page 7, line 193: "where Z is matrix with entires" $\longrightarrow$ "where Z is the matrix with entries"

17. Page 7: It would be worth referring to the monograph by Peyré and Cuturi (Peyré and Cuturi, 2019) on optimal transport (in particular section 4), since it is very well done and freely available.

18. Page 8, line 8, "ensemble Kalman inversion (EKI) is one of the widely used algorithm." it has other (better known) names such as Randomized Maximum Likelihood (RML) and Randomize-Then-Optimize. Its sequential variant is known as the very well known EDA (Ensemble of Data Assimilation) in the numerical weather prediction/data assimilation community.

19. Page 8, line 218: "By implementing a sequential observation update of Whitaker et al. (2008),": what do you mean by this statement?

20. Page 8, line 224: "is remarkable robust" ⟶ "is remarkably robust"

21. Page 9, Eqs.(14,15): I don't understand the intermediate member of both equations. The $\beta$ or $1 - \beta$ should be powers of $g$, and not multiply $g$. Or is this a notation? What did I miss?

22. Page 9, line 238-239: "This ansatz can also be understood as using the EKI as an more elaborate proposal density for the importance sampling step within SMC.": Using RML as a proposal density was already proposed and tested by Oliver et al. (1996).

23. Page 9, line 244-245: Are (x,y) horizontal dimensions or is y the depth? I believe it is worth explaining.

24. Page 10, 258: "$\delta$ Dirac function" ⟶ "$\delta$ the Dirac function"

25. Page 10, line 262: "We assume log permeability for" ⟶ "We assume that the log permeability for"

26. Page 10, line 267: Ok, but which type of solver did you use? (multigrid, linear algebra solver, etc.)

27. Page 11, line 272: "The grid dimension is 70" $\longrightarrow$ "The grid dimension is N=70"

28. Page 11, line 277: "The grid dimension is 50" $\longrightarrow$ "The grid dimension is N=50"

29. Page 11, lines 293-295: "Such a small noise makes the data assimilation problem hard to solve, since the likelihood is very peaked and a non-iterative data assimilation approach fails.": the explanation is very unclear to me. Please clarify.

30. Page 12, line 301: "An MCMC solution was obtained by combining 50 independent chains each of length $10^6$": this contradicts to some extent the statement made about its serial nature in the introduction.

31. Page 12, line 223: $9$ observation seem too few, are they? Your experiments might rely too much on the prior. I guess for reservoir or hydrological applications, there are indeed just a a few points, but they are many measurements over time at the same well.

32. Page 12, line 323: "distributed observations. which are displayed" $\longrightarrow$ "distributed observations, which are displayed"

33. Page 13, Figure 2: please add a label ($\beta$) to the x-axis.

34. Page 13, Figure 2: At $\beta = 0$ there is quite a discrepancy between the N=100 and the N=500 experiments. This could show that EKI (alone) is not working very well here. Moreover, quite often, the whiskers for N=100 and N=500 have no overlap. We would expect some overlap, would we? Do you have an interpretation?

35. page 13, Figure 3: By "Optimal" in the labelling of the panels, do you mean optimal transport, or something else?

36. page 14, Figure 4: Now, there is some consistent overlap between the N=100 and N=500 experiments, because, I guess, of the limited number of parameters (the curse of dimensionality is avoided in this case).

37. Page 14, line 333: "is lowest though" ⟶ "is lowest although".

38. Page 15, lines 362-363: "This makes the proposed method a promising option for the high dimensional nonlinear problems one is typically faced with in geophysical applications.". Your problem do not have time dependence (does it?) which often makes many geophysical applications (like meteorology and ocean forecasting) very difficult. So you could mitigate that statement.

39. Page 16, Figure 6: What about the prior? How does it compare to the posterior?

40. Page 17: "approach provides all the desirable properties required to obtain robust and highly accurate approximate solutions of nonlinear high dimensional Bayesian inference problems.": You cannot really make such a bold statement from one (however nice) example. Please mitigate your statement.

41. General question which is worth discussing a bit: In practice, how fast is the Sinkhorm numerical solution compared to the exact optimal transport?

**References**

Bardsley, J.M., Solonen, A., Haario, H., Laine, M., 2014. Randomize-then-optimize: A method for sampling from posterior distributions in nonlinear inverse problems. SIAM J. Sci. Comput. 36, A1895–A1910.

Liu, Y., Haussaire, J.M., Bocquet, M., Roustan, Y., Saunier, O., Mathieu, A., 2017. Uncertainty quantification of pollutant source retrieval: comparison of Bayesian methods with application to the Chernobyl and Fukushima-Daiichi accidental releases of radionuclides. Q. J. R. Meteorol. Soc. 143, 2886–2901. doi:10.1002/qj.3138.

Oliver, D.S., He, N., Reynolds, A.C., 1996. Conditioning permeability fields to pressure data, in: ECMOR V-5th European Conference on the Mathematics of Oil Recovery, pp. 259–269.

Peyré, G., Cuturi, M., 2019. Computational optimal transport: With applications to data science. Foundations and Trends® in Machine Learning 11, 355–607. doi:10.1561/2200000073.

---

## Author Comment (AC1) · 9 Oct 2020

Sangeetika Ruchi[1], Svetlana Dubinkina[1], and Jana de Wiljes[2]

[1]Centrum Wiskunde & Informatica, P.O. Box 94079, 1098 XG Amsterdam, the Netherlands
[2]University of Potsdam, Karl-Liebknecht-Str. 24/25, 14476, Potsdam, Germany

**Correspondence:** Jana de Wiljes (wiljes@uni-potsdam.de)

We would like to thank Femke Vossepoel and Marc Bocquet for carefully reading the manuscript and for their insightful comments and suggestions that definitely improved our article.

**Point-by-point answer to the comments by Femke Vossepoel**

Specific comments:

1. Terminology and description of example: As this paper could be of particular use to practitioners in the reservoir-engineering domain, I would encourage the authors to make the text more accessible to those. This could be done by changing or clarifying the use of certain terms and adding key references to explain the methods. For example, in reservoir engineering, the term Ensemble Kalman Filter is more commonly used than the term Ensemble Kalman Inversion; adding a number of key publications on this method and derived methods would help to set the scene and provide the reader with further background information. Also, those using data assimilation in practical applications will be interested in the actual values of the properties, and less likely to work on dimensionless problems. Relating the symbols to physical quantities would make this manuscript more accessible and relevant to them.

   **Response:** Thank you for raising this important point. It is crucial for us to reach practitioners and make the paper accessible to a wider audience and we have taken your suggestions into account. For instance we now use the term *Ensemble Kalman filter* instead of *Ensemble Kalman Inversion* as it is much more prevalent in the applied communities. Further we added a short discussion on the different terms and methods including randomised maximum likelihood, which is very popular with practitioners, and how they relate to each other.

2. Presentation of the methodologies: The mathematical rigour and expertise of the authors would allow them to not only compare the performance of the methods in an empirical sense, but also place them in the overall framework of data-assimilation methods for parameter estimation. The manner

in which the hybrid EKI-TETPF method is presented, is presented as a particle filter with several "fixes" (namely a) tempering, b) a Sinkhorn approximation, and c) an EKI proposal).

Can the authors think of a way to present the methods from a holistic viewpoint, making clear that these "fixes" are essential ingredients of the methods in order to perform a consistent and also effective parameter estimation? The abstract reads "Gaussian approximations [....] often produce astonishingly accurate estimations despite the inherently wrong underlying assumptions." Can you discuss more explicitly how the assumption of Gaussianity affects the outcome, perhaps by illustrating how non-Gaussian the distributions really are, or how the different methods deal with non- Gaussianity and/or non-linearity?

**Response:** The chosen presentation via particle filters (or Sequential Monte Carlo) allowed us to introduce all considered methods within one overarching family of filters. In order to make clear which techniques are standalone methods and which fixes they required to make the feasible in a challenging setting, we added the following text:

> In the following we will introduce a range of methods that can be employed to estimate solutions to the presented inverse problem under the overarching mantel of tempered Sequential Monte Carlo filters. Alongside these methods we will also proposed several important add-on tools required to achieve feasibility and higher accuracy in high-dimensional non-linear settings.

The sentence on Gaussian approximations has been adjusted (see our response to comment 3).

Technical corrections (language, minor items)

1. Please pay attention to the use of hyphens in compound modifiers. For example, the title could read "Application of ensemble-transform data-assimilation methods for parameter estimation in nonlinear problems". Other places where this would help: "high-dimensional problems", "entropy-inspired", "highly-correlated samples", "an easy-to-sample form|', etc.

**Response:** Thank you this suggestion. We now use the hyphens high-dimensional, entropy-inspired, highly-correlated samples and easy-to-sample within the manuscript in order to increase readability.

The original title was changed to "Fast hybrid tempered ensemble transform filter formulation for Bayesian elliptical problems via Sinkhorn approximation" and we preferred not to have the hyphen ensemble-transform.

2. The term "ensemble Kalman inversion" is used to a method that is known by many as "ensemble Kalman filtering". I suggest to clarify that EKI is used as equivalent to the ensemble Kalman filter. Page 2, line 38 and/or in the paragraph starting on p.8, line 201: suggest to add one of the key references for ensemble Kalman inversion or ensemble Kalman filtering, so readers can find out more about the method.

**Response:** We have changed Ensemble Kalman Inversion to Ensemble Kalman filter everywhere in the manuscript. Furthermore, we now mention that the method is known under different names

in different communities: randomized maximum likelihood, multiple data assimilation, ensemble of data assimilation, ensemble Kalman inversion. The following text has been added to the manuscript:

> "As a side remark, EnKF was originally proposed for estimating a dynamical state of a chaotic system (e.g., Burgers et al., 1998). It was latter shown by Anderson (2001) that EnKF can be used for parameter estimation by introducing a trivial dynamics to the unknown static parameter. We note that EnKF is well known under different names in different scientific communities. In the reservoir community it is Ensemble Randomized Maximum Likelihood (Chen and Oliver, 2012), multiple data assimilation (Emerick and Reynolds, 2013), and Randomize-Then-Optimize (Bardsley et al., 2014). In the numerical weather prediction community, it falls under a large umbrella of Ensemble of Data Assimilation, see Carrassi et al. (2018) for a recent review. In the inverse problem community, it is ensemble Kalman inversion (Chada et al., 2018)."

3. Page 1, line 3: abstract: "inherently wrong": the Gaussian assumptions are not always wrong, so suggest to reformulate: "despite the simplifying assumptions" or something along these lines. Alternatively, demonstrate in the manuscript that these assumptions are actually wrong.

   **Response:** We have changed "inherently wrong" to "despite the simplifying assumptions".

4. Page 2, line 55: the number of required intermediate steps and the efficiency of ETPF still depends on it. What does *it* refer to?

   **Response:** Here *it* refers to the dependence on the initialisation. The corresponding text in the revised manuscript is "Although tempering restrains any sharp fail in the importance sampling step due to a poor initial ensemble selection, the number of required intermediate steps and the efficiency of ETPF still depends on the initialisation."

5. Page 5, line 118: Crank-Nicholson pcn-MCMC: explain what pcn means here.

   **Response:** pcn-MCMC means the preconditioned Crank-Nicolson MCMC.

6. Page 5 line 130: the scalar theta -> the scalar theta in Equation 5.

   **Response:** Thank you, we fixed it.

7. Page 6, line 152: where the minimum is compute -> where the minimum is. computed

   **Response:** Thank you, we fixed it.

8. Page 7 line 181 One the other hand -> on the other hand.

   **Response:** Thank you, we fixed it.

9. Page 8, line 204: estimation of posterior -> estimation of the posterior.

   **Response:** Thank you, we fixed it.

10. Page 8, line 205-215: make sure to list and clarify all symbols used.

**Response:** Please see the clarification (the same text is added to the revised manuscript):

The intermediate measures $\{\mu_t\}_{t=0}^T$ are approximated by Gaussian distributed variables with empirical mean $m_t$ and empirical variance $\mathbf{C}_t$. Empirical mean $m_{t-1}$ and empirical covariance $\mathbf{C}_{t-1}$ are defined in terms of $\{u_{t-1,i}\}_{i=1}^M$ as following

$$m_{t-1} = \frac{1}{M}\sum_{i=1}^M u_{t-1,i}, \qquad \mathbf{C}_{t-1} = \frac{1}{M-1}\sum_{i=1}^M (u_{t-1,i} - m_{t-1}) \otimes (u_{t-1,i} - m_{t-1}),$$

where $\otimes$ denotes Kroneker product. Then the mean and the covariance are updated as

$$m_t = m_{t-1} + \mathbf{C}_{t-1}^{\mathrm{uF}}(\mathbf{C}_{t-1}^{\mathrm{FF}} + \Delta_t \mathbf{R})^{-1}(y_{\mathrm{obs}} - \overline{F}_{t-1}) \qquad \text{and} \qquad \mathbf{C}_t = \mathbf{C}_{t-1} - \mathbf{C}_{t-1}^{\mathrm{uF}}(\mathbf{C}_{t-1}^{\mathrm{FF}} + \Delta_t \mathbf{R})^{-1}(\mathbf{C}_{t-1}^{\mathrm{uF}})',$$

respectively. Here $\prime$ denotes the transpose,

$$\mathbf{C}_{t-1}^{\mathrm{uF}} = \frac{1}{M-1}\sum_{i=1}^M (u_{t-1,i} - m_{t-1}) \otimes (F(u_{t-1,i}) - \overline{F}_{t-1}), \quad \mathbf{C}_{t-1}^{\mathrm{FF}} = \frac{1}{M-1}\sum_{i=1}^M [F(u_{t-1,i}) - \overline{F}_{t-1}] \otimes [F(u_{t-1,i}) - \overline{F}_{t-1}],$$

$$\overline{F}_{t-1} = \frac{1}{M}\sum_{i=1}^M F(u_{t-1,i}), \qquad \text{and} \qquad \Delta_t = \frac{1}{\phi_t - \phi_{t-1}}.$$

We recall that the nonlinear forward problem is $y = F(u)$, the observation $y_{\mathrm{obs}}$ has a Gaussian observation noise with zero mean and the covariance matrix $\mathbf{R}$, and $\phi_t$ is a temperature associated with the measure $\mu_t$.

11. Page 9, line 232: make clear how to choose beta.

**Response:** We agree that the choice of $\beta$ needs to be discussed. For our concrete numerical setting we added the following information in the numerical section:

"Note that $\beta \in [0,1]$ and should be tuned according to underlying forward operator. "

We later also address the choice of $\beta$ in more detail in the conclusion (please see our response to comment 24).

12. Page 9, line 239: EKI as an more elaborate -> as a more elaborate.

**Response:** Thank you, we fixed it.

13. Page 9, line 240: Computational complexity: the estimates of computational complexity of the various methods is very useful. I suggest to include a table that illustrates the computational complexity of all methods/variations and include a few sentences on this in the 'Conclusions' part.

**Response:** Thank you for this suggestion. We added the following table (Table **??**) to the appendix and now elaborate on the computational complexity in the conclusion.

14. Page 9, line 244: the example is dimensionless. Suggest to relate this to an example in which you list a number of typical values. You can then also mention that channels such as shown in Fig 1 can be found in fluvial rock formations that form aquifers or reservoirs.

| Algorithm | Complexity |
|---|---|
| TETPF | $\mathcal{O}[T(M\mathcal{C} + M^3 \log M + \tau_{\max} M\mathcal{C})]$ |
| TESPF | $\mathcal{O}[T(M\mathcal{C} + M^2 C(\alpha) + \tau_{\max} M\mathcal{C})]$ |
| EnKF | $\mathcal{O}[T(M\mathcal{C} + \kappa^2 n + \tau_{\max} M\mathcal{C})]$ |
| Hybrid EnKF-TETPF | $\mathcal{O}[T(M\mathcal{C} + \kappa^2 n + M\mathcal{C} + M^3 \log M + \tau_{\max} M\mathcal{C})]$ |
| Hybrid EnKF-TESPF | $\mathcal{O}[T(M\mathcal{C} + \kappa^2 n + M\mathcal{C} + M^2 C(\alpha) + \tau_{\max} M\mathcal{C})]$ |
| Forward model | $\mathcal{O}(M\mathcal{C})$ |
| pcn-MCMC mutation | $\mathcal{O}(\tau_{\max} M\mathcal{C})$ |
| FastEMD | $\mathcal{O}(M^3 \log M)$ |
| Sinkhorn approximation | $\mathcal{O}(M^2 C(\alpha))$ |

**Table 1.** The table provides an overview of the computational complexity of all the algorithms considered in the manuscript.

**Response:** We relate now to the paper by Zovi et al. (2017): "We note that a single-phase Darcy flow
25    model, though not a steady-state, is widely used to model the flow in a subsurface aquifer and to
infer uncertain permeability using data assimilation. For example, Zovi et al. (2017) used an EnKF
to infer permeability of an existing aquifer located in North-East Italy. The area of this aquifer is 2.7
km$^2$ and exhibits several channels, such as the one depicted in Fig. 1. There a size of a computational
cell was ranging from 2 m (near wells) to 20 m away from the wells."

30   15. Page 9, line 247: please make clear what physical variable (rate, pressure) the source term represents.

     **Response:** The source term $f$ accounts for groundwater recharge. This text is added to the revised
manuscript.

16. Page 9 line 267: on an $N \times N$ grid: a potential user would like to know what is the scale, and spatial
dimension. Suggest to give the value of $N$ earlier than you do now (on page 10, line 273).

35     **Response:** The details of the numerical approximation are now given right after the continuous
formulation.

17. Page 10, line 255: the choice of P for parameterisation is not very practical, as you are also using this
letter for pressure.

     **Response:** It is changed to F for parameterisation.

40   18. Page 11, line 285: please make clear what property is being observed.

     **Response:** We observe the pressure at a few grid points. We have changed the text accordingly.

19. Page 12, line 310: we plot box plot -> we plot a box plot; using Sinkhorn approximation -> using a
Sinkhorn approximation.

     **Response:** Thank you, we fixed it.

20. Page 12, line 313: TESPF outperforms: has a lower RMSE? Is smoother? How do you define a good performance?

**Response:** TESPF outperforms TETPF as the RMSE error is lower. The corresponding text is added to the revised manuscript.

21. Page 12 line 320: estimate well mot -> estimate well not.

**Response:** Thank you, we fixed it.

22. Page 13, figure 2: it is good to see the box plots for permeability, I would have liked to also see this for rate (observed state variable).

**Response:** We compute the pressure of the mean log permeability and plot a corresponding box plot for the RMSE in Figure **??**.

First, we see that the smaller is the $\beta$, the smaller is the error. Next, we see that at the large ensemble size $M = 500$ the optimal transport resampling (shown in Figure **??**(b)) outperforms the Sinkhorn approximation (shown in Figure 1(a)) in terms of smaller error. These two conclusions hold for the mean log permeability shown in Figure **??**. The difference is that at the smaller ensemble size $M = 100$ the optimal transport resampling (shown in Figure 1(b)) outperforms the Sinkhorn approximation (shown in Figure **??**(a)) in terms of smaller error for *the pressure*. However, for the *log permeability* it is the Sinkhorn approximation (shown in Figure **??**(a)) that outperforms the optimal transport resampling (shown in Figure **??**(b)) for $\beta \geq 0.6$ in terms of smaller error. We attribute this inconsistency to the cancellation of errors when computing the pressure of the mean log permeability.

[Figure]

**Figure 1.** Application to F1 parameterization: using Sinkhorn approximation (a) and optimal transport resampling (b). Box plot over 20 independent simulations of RMSE of the pressure of the mean log permeability. X-axis is for the hybrid parameter, where $\beta = 0$ corresponds to EnKF and $\beta = 1$ to TET(S)PF. Ensemble size $M = 100$ is shown in red, and $M = 500$ in green. Central mark is the median, edges of the box are the 25th and 75th percentiles, whiskers extend to the most extreme datapoints, and crosses are outliers.

23. Page 14 figure 4: please label the x axis (it is described in the caption but would be good to see in the figure, too).

**Response:** Thank you, we fixed it.

[Figure]

**Figure 2.** Application to F1 parameterization: using Sinkhorn approximation (a) and optimal transport resampling (b). Box plot over 20 independent simulations of RMSE of mean log permeability. X-axis is for the hybrid parameter, where $\beta = 0$ corresponds to EnKF and $\beta = 1$ to TET(S)PF. Ensemble size $M = 100$ is shown in red, and $M = 500$ in green. Central mark is the median, edges of the box are the 25th and 75th percentiles, whiskers extend to the most extreme datapoints, and crosses are outliers.

24. Page 14, line 332: these are very interesting results. I would value a discussion on how to find the best $\beta$ value in a realistic application of the hybrid method. In a synthetic case, this value can be determined, but how would you deal with this when assimilating real data? This discussion could be added in "conclusions: (page 17).

    **Response:** Thank you for raising this point, we added the following discussion to the conclusion:

    Note that we have considered a synthetic case, where the truth is available, and thus chose $\beta$ in terms of accuracy of an estimate. However, in a realistic application the truth is not provided. In the context of state estimation with an underlying dynamical system it has been suggested to adaptively change the hybrid parameter with respect to the effective sample size. As the tempering scheme is already changed according to the effective sample size this ansatz would require to define the interplay between the two tuning variables. An ad-hoc choice for $\beta$ could be $0.2$ or $0.3$. This is motivated by the fact that the particle filter is too unstable in high dimensions and it is therefore sensible to use a tuning parameter prioritising the EnKF. The ad-hoc choice is supported by the numerical results in Section 3 and in Acevedo et al. (2017); de Wiljes et al. (2020) in the context of state-estimation.

25. Page 14 line 344: we plot box plot -> we plot a box plot (or "the box plot shows...").

    **Response:** Thank you, we fixed it.

26. Page 15, line 363: the application that you show, would be referred to as a "reservoir engineering" application, or a "hydrological" application, not as a "geophysical application". (In oil- and gas industry, reservoir engineering is about flow in porous media, while geophysics is about the use of seismic and other geophysical data and propagation of sound waves. In hydrology, permeability is usually replaced by hydraulic conductivity, so by using permeability your example would be more familiar to those working in reservoir engineering.)

**Response:** We are very grateful for you comment as we belief that these specifics are crucial to make our manuscript comprehensible for readers from the various community. Thank you also for taking the time to clarify the specifications of the fields. We changed geophysical application to reservoir engineering.

**Point-by-point answer to the comments by Marc Bocquet**

1. Page 1: I believe that the title of the paper is too generic, not specific enough. It could suit dozens of papers already published. I strongly suggest that you revise it. I understand that this is not easy since you use a large collection of methods. Although quick to amend, I believe this point is problematic for the visibility/identification of the paper.

   **Response:** We agree and have changed the title to "Fast hybrid tempered ensemble transform filter formulation for Bayesian elliptical problems via Sinkhorn approximation"

2. Page 1, line 2: "Kalman inversion" is not a widespread terminology, "randomized maximum like-lihood" is better known, even beyond the reservoir community. See Oliver et al. (1996) and many references since then.

   **Response:** We have changed ensemble Kalman inversion to a better known ensemble Kalman filter. Furthermore, we added text about different name in different scientific communities. Namely:

   "As a side remark, EnKF was originally proposed for estimating a dynamical state of a chaotic system (e.g., Burgers et al., 1998). It was latter shown by Anderson (2001) that EnKF can be used for parameter estimation by introducing a trivial dynamics to the unknown static parameter. We note that EnKF is well known under different names in different scientific communities. In the reservoir community it is Ensemble Randomized Maximum Likelihood (Chen and Oliver, 2012), multiple data assimilation (Emerick and Reynolds, 2013), and Randomize-Then-Optimize (Bardsley et al., 2014). In the numerical weather prediction community, it falls under a large umbrella of Ensemble of Data Assim-ilation, see Carrassi et al. (2018) for a recent review. In the inverse problem community, it is ensemble Kalman inversion (Chada et al., 2018)."

3. Page 1, line 4: "of the associated statistics.": I am not sure to get what you mean.

   **Response:** We mean that we can go beyond Gaussian approximations even with ensemble Kalman filter. We have adjusted the text accordingly:

   Yet there is a lot of room for improvement specifically regarding a correct approximation of a non-Gaussian posterior distribution.

4. page 1, line 18, "a just approximation": do you mean a "correct approximation"?

   **Response:** Yes, we have adjusted the text correspondingly.

5. page 2, line 28, "The main drawback of MCMC is that this approach is not parallelizable.": You know that there are parallel (multiple tries) versions of MCMCs. It actually seems that you are yourself using multiple parallel MCMCs. So I believe you should mitigate that statement.

**Response:** Indeed, we have omitted this statement. Instead we emphasise that MCMC samples are highly correlated. The following text has been added:

> Typically, MCMC methods provide highly correlated samples. Therefore, in order to sample the posterior correctly MCMC requires a long chain, especially in the case of a multimodal or a peaked distribution. A peaked posterior is associated with very accurate observations.

6. Page 2, line 41-43: "However for nonlinear problems, Ernst et al. (2015); Evensen (2018) showed that in the large ensemble size limit an EKI approximation is not consistent with the Bayesian approximation.": To the best of my knowledge this is has been pointed out first by Oliver et al. (1996). The mathematical problem has also been clearly defined by Bardsley et al. (2014), and nicely named 'Randomize-Then-Optimize". There is also a recent discussion on the issue in Liu et al. (2017), p. 2894.

**Response:** We have now included these references. Namely:

> However for nonlinear problems, it has been shown by Oliver et al. (1996); Bardsley et al. (2014); Ernst et al. (2015); Liu et al. (2017) that an EnKF approximation is not consistent with the Bayesian approximation.

7. Page 2, line 57-58: Yes, but you should at this point mention here that the idea originates from the optimal transport community, and that it is by now widespread.

**Response:** Thank you for pointing this out. The sentence is indeed misleading. Additionally we add a more extensive literature survey of hybrid filters as we felt we did not do it justice before. The following text address both issues and is now added to the revised version of the manuscript:

> The lack of robustness in high dimensions can be addressed via a hybrid approach that combines a Gaussian approximation with a particle filter approximation (e.g., Santitissadeekorn and Jones, 2015). Different algorithms are created by Frei and Künsch (2013); Stordal et al. (2011), for example. In this paper, we adapt a hybrid approach of Chustagulprom et al. (2016) that uses EnKF as a proposal step for ETPF with a tuning parameter. Furthermore, it is well established that the computational complexity of solving an optimal transport problem can be significantly reduced via a Sinkhorn approximation by Cuturi (2013). This ansatz has been been implemented for the ETPF in Acevedo et al. (2017).

8. Page 3, line 73: even though obvious, it would be better to mention explicitly that $\mathcal{N}$ is the Gaussian distribution.

**Response:** Thank you, we fixed it.

9. Page 4, line 114: "Mutation" is applied mathematics Pierre Del Moral's terminology. You could briefly explain what it corresponds to in the geophysics particle filter community (rejuvenation?)

   **Response:** We have added the following text:

   > In the framework of particle filtering for dynamical systems, ensemble perturbation is achieved by rejuvenation, when ensemble members of the posterior measure are perturbed with a random noise sampled from a Gaussian distribution with zero mean and a covariance matrix of the prior measure. The covariance matrix of the ensemble is inflated and no acceptance step is performed due to the associated high computational costs for a dynamical system.

   > Since we consider a static inverse problem, for ensemble perturbation we employ a Metropolis–Hastings method (thus we mutate samples) but with a proposal that speeds up an MCMC method for estimating a high-dimensional parameter.

10. Page 4, line 122: "we use random walk" → "we use the following random walk".

    **Response:** Thank you, we fixed it.

11. Page 5, line 135: "where C is computational cost of a forward model F" → "where C is the computational cost of the forward model F".

    **Response:** Thank you, we fixed it.

12. Page 6, line 136: 'is not effected"→"is not affected".

    **Response:** Thank you, we fixed it.

13. Page 6, line 150: "we seak" → "we seek".

    **Response:** Thank you, we fixed it.

14. Page 6, line 160, Eq.(10): What is the definition of the norm of the random variables used in this equation?

    **Response:** Thank you for pointing this out. We have changed the equation to the following:

$$
\omega^* = \arg\inf \left\{ \int_{\tilde{\mathcal{U}} \times \tilde{\mathcal{V}}} \|\boldsymbol{u} - \tilde{\boldsymbol{u}}\|^2 d\omega(\boldsymbol{u}, \tilde{\boldsymbol{u}}) : \quad \omega \in \prod(\mu, \nu) \right\}. \tag{1}
$$

15. Page 7, line 181: "One the other hand"→"On the other hand".

    **Response:** Thank you, we fixed it.

16. Page 7, line 193: "where Z is matrix with entires"→ "where Z is the matrix with entries".

    **Response:** Thank you, we fixed it.

17. Page 7: It would be worth referring to the monograph by Peyré and Cuturi (Peyré and Cuturi, 2019) on optimal transport (in particular section 4), since it is very well done and freely available.

    **Response:** Thank you for the suggestion. We now refer the reader to the monograph.

18. Page 8, line 8, "ensemble Kalman inversion (EKI) is one of the widely used algorithm." it has other (better known) names such as Randomized Maximum Likelihood (RML) and Randomize-Then-Optimize. Its sequential variant is known as the very well known EDA (Ensemble of Data Assimilation) in the numerical weather prediction/data assimilation community.

    **Response:** We have addressed this in the revised manuscript. Please see our response to comment 2. above.

19. Page 8, line 218: "By implementing a sequential observation update of Whitaker et al. (2008),": what do you mean by this statement?

    **Response:** We have omitted this statement, since it is irrelevant for a small number of observations. Instead, we state that

    The computational complexity of solving Eq.(13) is $\mathcal{O}(\kappa^2 n)$, where $n$ is the parameter space dimension, and $\kappa$ is the observation space dimension.

20. Page 8, line 224: "is remarkable robust" → "is remarkably robust".

    **Response:** Thank you, we fixed it.

21. Page 9, Eqs.(14,15): I don't understand the intermediate member of both equations. The $\beta$ or $1 - \beta$ should be powers of g, and not multiply g. Or is this a notation? What did I miss?

    **Response:** Thank you for pointing out this typo. Indeed, $\beta$ or $1 - \beta$ should be powers of $g$. It is now fixed in the revised manuscript.

22. Page 9, line 238-239: "This ansatz can also be understood as using the EKI as an more elaborate proposal density for the importance sampling step within SMC.": Using RML as a proposal density was already proposed and tested by Oliver et al. (1996).

    **Response:** Thank you for pointing out this reference, which we now add in the revised manuscript: "This ansatz can also be understood as using the EnKF as a more elaborate proposal density for the importance sampling step within SMC (e.g., Oliver et al., 1996)."

23. Page 9, line 244-245: Are $(x, y)$ horizontal dimensions or is $y$ the depth? I believe it is worth explaining.

    **Response:** $(x, y)$ are horizontal dimensions. We now add this in the revised manuscript.

24. Page 10, 258: "$\delta$ Dirac function" → "$\delta$ the Dirac function".

    **Response:** Thank you, we fixed it.

25. Page 10, line 262: "We assume log permeability for" → "We assume that the log permeability for".

    **Response:** Thank you, we fixed it.

26. Page 10, line 267: Ok, but which type of solver did you use? (multigrid, linear algebra solver, etc.)

    **Response:** We use a linear algebra solver (backslash operator in MATLAB). The corresponding text in now added to the revised version.

27. Page 11, line 272: "The grid dimension is 70" → "The grid dimension is $N = 70$".

    **Response:** Thank you, we fixed it.

28. Page 11, line 277: "The grid dimension is 50"→ "The grid dimension is $N = 50$".

    **Response:** Thank you, we fixed it.

29. Page 11, lines 293-295: "Such a small noise makes the data assimilation problem hard to solve, since the likelihood is very peaked and a non-iterative data assimilation approach fails.": the explanation is very unclear to me. Please clarify.

    **Response:** With such a small noise the likelihood is a peaked distribution. Therefore a non-iterative data assimilation approach requires a computationally unfeasible number of ensemble members to sample the posterior. This text is now added to the revised manuscript.

30. Page 12, line 301: "An MCMC solution was obtained by combining 50 independent chains each of length 106": this contradicts to some extent the statement made about its serial nature in the introduction.

    **Response:** We have omitted the statement about MCMC serial nature made earlier in the introduction.

31. Page 12, line 223: 9 observation seem too few, are they? Your experiments might rely too much on the prior. I guess for reservoir or hydrological applications, there are indeed just a few points, but they are many measurements over time at the same well.

    **Response:** This is a fair criticism. Fever observations allow for a multi-modal posterior. More observations (either due to more wells or due to measurements at a few wells but over some time interval) decrease the uncertainty resulting in a uni-modal posterior.

32. Page 12, line 323: "distributed observations. which are displayed" → "distributed observations, which are displayed"

    **Response:** Thank you, we fixed it.

33. Page 13, Figure 2: please add a label ($\beta$) to the x-axis.

    **Response:** Thank you, we fixed it.

34. Page 13, Figure 2: At $\beta = 0$ there is quite a discrepancy between the $M = 100$ and the $M = 500$ experiments. This could show that EKI (alone) is not working very well here. Moreover, quite often, the whiskers for $M = 100$ and $M = 500$ have no overlap. We would expect some overlap, would we? Do you have an interpretation?

**Response:**

A discrepancy between the $M = 100$ and the $M = 500$ experiments at $\beta = 0$ (thus EnKF alone) is related to the curse of dimensionality. The ensemble size $M = 100$ is too small to estimate an uncertain parameter of the dimension $10^3$ using 36 accurate observations. However, at the ensemble size $M = 500$ EnKF alone ($\beta = 0$) gives an excellent performance compared to any combination ($\beta > 0$).

We now add the above text to the revised manuscript.

Indeed, the overlap in the whiskers is to be expected. Therefore we have performed experiments with EnKF ($\beta = 0$) at the ensemble sizes $M = 200$ and $M = 400$. The box plot of the error is shown in Figure **??**. We see that as ensemble size increases the whiskers get close to each other and we see an overlap between $M = 400$ and $M = 500$.

[Figure]

**Figure 3.** Application to F1 parameterization: using Sinkhorn approximation. Box plot over 20 independent simulations of RMSE of mean log permeability. X-axis is for the hybrid parameter, where $\beta = 0$ corresponds to EnKF and $\beta = 1$ to TET(S)PF. Ensemble size $M = 100$ is shown in red, $M = 200$ in green, $M = 400$ in blue, and $M = 500$ in black. Central mark is the median, edges of the box are the 25th and 75th percentiles, whiskers extend to the most extreme datapoints, and crosses are outliers.

35. page 13, Figure 3: By "Optimal" in the labelling of the panels, do you mean optimal transport, or something else?

**Response:** We see the confusion. Indeed, by "Optimal" in the labelling of the panels we mean optimal transport. We now change it to "OT".

36. page 14, Figure 4: Now, there is some consistent overlap between the $M = 100$ and $M = 500$ experiments, because, I guess, of the limited number of parameters (the curse of dimensionality is avoided in this case).

**Response:** Indeed, in this numerical experiment we have less observations (only 9). The P2 parameterization has 5 geometrical parameters and of order $10^3$ permeability parameters.

37. Page 14, line 333: "is lowest though" $\rightarrow$ "is lowest although".

**Response:** Thank you, we fixed it.

38. Page 15, lines 362-363: "This makes the proposed method a promising option for the high dimensional nonlinear problems one is typically faced with in geophysical applications.". Your problem do not have time dependence (does it?) which often makes many geophysical applications (like meteorology and ocean forecasting) very difficult. So you could mitigate that statement.

**Response:** We agree that we need to be more specific in this sentence and replaced "geophysical applications" with "reservoir engineering".

39. Page 16, Figure 6: What about the prior? How does it compare to the posterior?

**Response:** The prior is uniform, namely $d^1 \sim U[0.3, \ 2.1]$, $d^2 \sim U[\pi/2, \ 6\pi]$, $d^3 \sim U[-\pi/2, \ \pi/2]$, $d^4 \sim U[0, \ 6]$, $d^5 \sim U[0.12, \ 4.2]$. We see that the posterior is not a uniform distribution.

40. Page 17: "approach provides all the desirable properties required to obtain robust and highly accurate approximate solutions of nonlinear high dimensional Bayesian inference problems.": You cannot really make such a bold statement from one (however nice) example. Please mitigate your statement.

**Response:** We have changed this statement to the following:

> "This suggests a hybrid approach has a great potential to obtain robust and highly-accurate approximate solutions of nonlinear high-dimensional Bayesian inference problems."

41. General question which is worth discussing a bit: In practice, how fast is the Sinkhorn numerical solution compared to the exact optimal transport?

**Response:** This is a good question. Yet it is difficult to provide a general answer as the computational complexity of the Sinkhorn approximation is highly dependent on the choice of the regularisation parameter $\alpha$, i.e, specifically $\mathcal{O}(M^2 C(\alpha))$. It is know that $C(\alpha)$ grows with $\alpha$ as the original transport problem is approached yet the associated computational complexity can vary considerably. Therefore it is important to find an acceptable trade-off between a good approximation and the improvement in computational complexity. This fact was not discussed in the manuscript and we now added the following discussion to the conclusions:

> "Note however that $C(\alpha)$ depends on the chosen regularisation and grows with $\alpha$. Therefore, one needs to balance between reducing computational time and finding a reasonable approximate solution of the original transport problem when choosing a value for $\alpha$."

---

## Editor Decision (ED1)

Dear Jana, Sangeetika and Svetlana,

Two referees have submitted their reports on the revised version of your paper. They are the same referees as for the previous version. Referee 1 (Femke Vossepoel) recommends acceptance of the paper as it is. Referee 2 (Marc Bocquet) also recommends acceptance, subject to a few modifications, in particular on the use you make of the expression *Ensemble Kalman filter* (*EnKF*).

I myself as Editor make below a number of suggestions for modifications, including one on the expression *Ensemble Kalman filter* (but not the same one as Marc Bocquet).

Please consider all these suggestions, and modify your paper accordingly if you agree with them (give an answer only if you disagree).

Feel free to get in touch with M. Bocquet or myself if you wish.

I thank you for having submitted your paper to *NPG*, and look forward to receiving the final version.

Olivier Talagrand
Editor, *NPG*

My Editor's suggestions (*line numbers are those of the version of the paper without explicit identification of the corrections from the previous version*)

Ll. 26-27, *Markov chain Monte Carlo (MCMC) methods* … Give references.

L. 40, *Ensemble Kalman filtering (EnKF) approximates only the first two moments of the posterior*, … EnKF produces an ensemble, from which estimates of higher order moments, such as skewness and kurtosis, can be obtained (although possibly with poor accuracy). I would suggest … *approximates primarily the first two moments* … (see also l. 223).

L. 43. You might add *Le Gland et al. (2011)* in the list of references you give there.

Ll. 49-50, *In the numerical weather prediction community, it falls under a large umbrella of Ensemble of Data Assimilation*, … In the numerical weather prediction community, the expression *Ensemble of Data Assimilation* does not refer to Ensemble Kalman Filter. It refers to an ensemble of variational assimilations, performed on independently perturbed data. Now you may wish to refer to all assimilation methods, whether of the EnKF type or not, which produce an ensemble of point estimates meant to sample the posterior probability distribution. If so, be more explicit. And see also the related comment by M. Bocquet.

L. 52, *In order to sample highly-correlated samples*, … What do you mean by *highly-correlated samples*?

L. 106, Do you mean *When an easy-to-sample **ensemble** from the prior $\mu_0$ does not*…

L. 108, Do you man … *or **lack** of accuracy of the observations*?

L. 120, … *to find $\phi_t$* (index)

L. 165, *A century later*, … Actually, it was more than a century later. I suggest *A century and a half later*, …

L. 171, Eq. (9), no meaning is given at this stage to $\boldsymbol{u}$ and $\boldsymbol{\tilde{u}}$

L. 207, … *that is constrained by* …

L. 216, $C(\alpha)$ does not seem to have been defined

L. 340, *The x-axis* … → *The horizontal axis* …

Same remark for captions of Figures 2, 4 and 7 (and maybe elsewhere). Do not mention an *x-axis* when there is no *x*.

L. 70, … *by Avecedo et al*.

L. 80, … *while another one leads to* …

L. 197, … *Algorithm 2*, I presume

REFERENCE

Le Gland, F., Monbet, V., and Tran, V.-D.: Large sample asymptotics for the ensemble Kalman Filter, *The Oxford handbook of nonlinear filtering*, Oxford University Press, 598–631, 2011.

---

## Author Response (AR3)

Dear Olivier,

thank you for your positive feedback and valuable comments. We fixed
all the issues you addressed and are happy to provide you with the
revised manuscript.

Kind regards
Sangeetika, Svetlana and Jana

Dear Marc,

thank you for taking the time to go through the manuscript again and
for giving valuable comments. We fixed the grammatical error and
went through the manuscript and added articles in front of the
various abbreviations for the different methods whenever they were
missing.

In particular we are grateful for your objection to our misuse of
the term EnKF and for your explanation with respect to that matter.
We debated for a while if we should use the common terminology of
the petroleum community to make the paper more accessible to people
in that field. Yet, as we did not want to adapt the previously
defined nomenclature of the ETPF and all of its variants
correspondingly, we decided to revert to the use of EKI.
Nevertheless, we do mention the family of iterative ensemble
smoothers in the manuscript and reference one of Geir's recent works
on it.

Kind regards
Sangeetika, Svetlana and Jana

[revised manuscript text omitted]